# A functionally conserved *STORR* gene fusion in *Papaver* species that diverged 16.8 million years ago

Theresa Catania[1,7], Yi Li[1,7], Thilo Winzer[1], David Harvey[1], Fergus Meade [1], Anna Caridi[1], Andrew Leech[2], Tony R. Larson [2], Zemin Ning[3], Jiyang Chang[4], Yves Van de Peer [4,5,6] & Ian A. Graham [1✉]

The *STORR* gene fusion event is considered essential for the evolution of the promorphinan/morphinan subclass of benzylisoquinoline alkaloids (BIAs) in opium poppy as the resulting bi-modular protein performs the isomerization of (*S*)- to (*R*)-reticuline essential for their biosynthesis. Here, we show that of the 12 *Papaver* species analysed those containing the *STORR* gene fusion also contain promorphinans/morphinans with one important exception. *P. californicum* encodes a functionally conserved *STORR* but does not produce promorphinans/morphinans. We also show that the gene fusion event occurred only once, between 16.8-24.1 million years ago before the separation of *P. californicum* from other Clade 2 *Papaver* species. The most abundant BIA in *P. californicum* is (*R*)-glaucine, a member of the aporphine subclass of BIAs, raising the possibility that STORR, once evolved, contributes to the biosynthesis of more than just the promorphinan/morphinan subclass of BIAs in the Papaveraceae.

[1] Centre for Novel Agricultural Products, Department of Biology, University of York, York YO10 5DD, UK. [2] Bioscience Technology Facility, Department of Biology, University of York, York YO10 5DD, UK. [3] The Wellcome Sanger Institute, Wellcome Genome Campus, Hinxton, Cambridge CB10 1SA, UK. [4] Department of Plant Biotechnology and Bioinformatics, Ghent University, and VIB Centre for Plant Systems Biology, 9052 Ghent, Belgium. [5] Centre for Microbial Ecology and Genomics, Department of Biochemistry, Genetics and Microbiology, University of Pretoria, Pretoria 0028, South Africa. [6] College of Horticulture, Academy for Advanced Interdisciplinary Studies, Nanjing Agricultural University, Nanjing, China. [7] These authors contributed equally: Theresa Catania, Yi Li. ✉email: ian.graham@york.ac.uk

The benzylisoquinoline alkaloids or BIA's represent a structurally diverse group predominantly identified in the order Ranunculales[1,2]. The naturally synthesised morphinans thebaine, oripavine, codeine and morphine are part of the BIA class of alkaloids, with morphine renowned for its powerful analgesic properties. They are naturally synthesised in the genus *Papaver* and are currently commercially produced in opium poppy, *Papaver somniferum*, from the Papaveraceae family. The commercial importance of opium poppy has led to its use as a model species for research into the biosynthetic pathway for morphinan production[2–5]. The common precursor and central branch point in the pathway for production of the many structurally distinct sub-classes of BIAs in the Ranunculales including morphinan, proto-berberine, phthalideisoquinoline and benzophenanthridine is the 1-benzylisoquinoline alkaloid, (S)-reticuline[6] (Fig. 1a). The gateway reaction leading to morphinan biosynthesis is catalysed by the STORR protein. Composed of P450 and oxidoreductase modules, this fused protein completes the epimerization of (S)- to (R)-reti-culine, the first step in the morphinan pathway[3,4,7]. *STORR* is clustered with four other genes involved in synthesis of the first morphinan, thebaine and ten genes involved in synthesis of the phthalideisoquinoline, noscapine, which together make up the BIA gene cluster in opium poppy[5].

Advances in genome sequencing technology and assembly offers the opportunity to compare the genome organisation of related species and provide insight into the role of events such as gene fusion and gene clustering in the evolution of specialized metabolites[8–14]. Here, we use such an approach together with transcriptomics, metabolomics and gene function analysis to determine the evolutionary sequence of events leading to the clustering of the genes encoding the *STORR* modules, the fusion of these genes to form a functional *STORR* and the clustering of the other four genes involved in morphinan production across a number of *Papaver* species.

## Results

**Metabolite and transcriptomic analysis of *Papaver* species.** The epimerization of (S)- to (R)-reticuline is followed by a sequence of conversions from (R)-reticuline to salutaridine, salutaridinol, salutaridinol-7-O-acetate and thebaine catalysed by the products of *SALSYN*[15], *SALR*[16], *SALAT*[17] and *THS*[18], respectively (Fig. 1a). To investigate the presence of the *STORR* gene fusion across the *Papaver* genus we selected nine other *Papaver* species that pro-vided good taxonomic coverage (Table 1; Fig. 1b; Supplementary Table 1), and analysed these alongside the reported metabolite and whole genome assembly data of opium poppy[5,14], *P. rhoeas* and *P. setigerum*[14].

To determine the presence of promorphinan and morphinan compounds and related genes in the 11 species metabolite profiling of latex from juvenile plants and capsule material post-

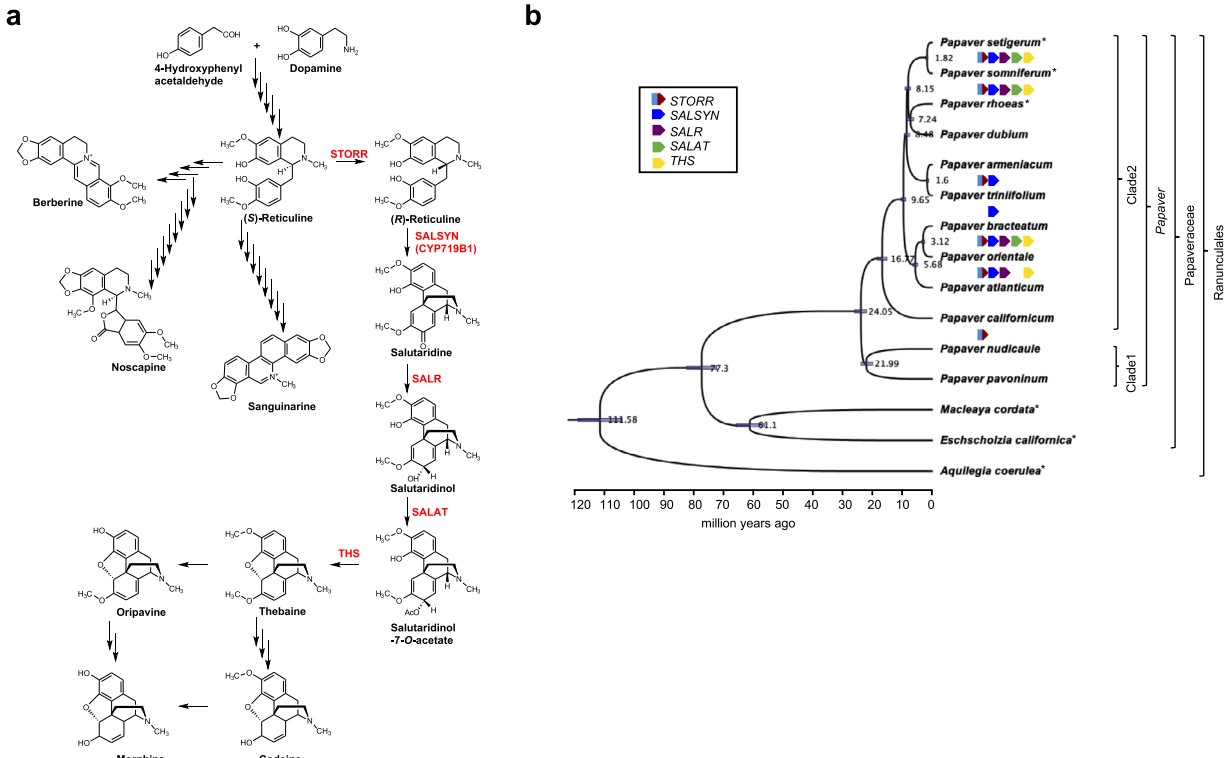

**Fig. 1 Metabolite and transcriptomic analysis to investigate *STORR* and its distribution in *Papaver* species. a** Schematic of the benzylisoquinoline pathway and enzymes in Opium Poppy with a focus on morphinan production. (S)-reticuline is the central branch point of BIA metabolism in opium poppy for the production of structurally distinct compounds including the morphinans plus noscapine, berberine and sanguinarine. Conversion of (S)- to (R)-reticuline by *STORR* represents the first committed step in biosynthesis of the promorphinans (salutaridine, Salutaridinol and salutaridinol-7-O-acetate) and morphinans (thebaine, oripavine, codeine and morphine). Compound names are in black and enzymes specific to the promorphinan and morphinan pathway are in red. **b** Species phylogeny inferred by Bayesian inference species tree using eight single copy conserved ortholog sequences. Phylogeny and divergence dates are estimated using BEAST2, which are shown on the nodes. Light-blue bars at the nodes indicate the range with 95% highest posterior density. Taxonomy groupings of the species are indicated and the twelve *Papaver* species are placed into two different clades (Clade 1 and Clade 2) as described by Carolan et al.[20]. Species highlighted with an asterisk indicate all gene sequences were identified and retrieved from the corresponding annotated genomes whereas for the remaining nine *Papaver* species, transcriptomic datasets were used. Presence of the *STORR* and the four promorphinan genes are shown under species names (Supplementary Data 2). Source data are provided as a Source Data file.

**Table 1 Metabolite analysis for promorphinan and morphinan compounds in 12 *Papaver* species\*.**

| Species | Reticuline | Promorphinan | | | Morphinan | | | |
| --- | --- | --- | --- | --- | --- | --- | --- | --- |
| | | Salutaridine | salutaridinol | Salutaridinol-7-O-acetate | Thebaine | Oripavine | Codeine | Morphine |
| *Papaver setigerum* | ND | ND | ND | ND | + | ND | + | + |
| *Papaver somniferum* | + | ND | ND | ND | + | ND | + | + |
| *Papaver rhoeas* | ND | ND | ND | ND | ND | ND | ND | ND |
| *Papaver dubium* | ND | ND | ND | ND | ND | ND | ND | ND |
| *Papaver armeniacum* | + | + | ND | ND | ND | ND | ND | ND |
| *Papaver triniifolium* | + | + | ND | ND | ND | ND | ND | ND |
| *Papaver bracteatum* | + | + | ND | ND | + | ND | ND | ND |
| *Papaver orientale* | + | + | ND | ND | + | + | ND | ND |
| *Papaver atlanticum* | + | ND | ND | ND | ND | ND | ND | ND |
| *Papaver californicum* | + | ND | ND | ND | ND | ND | ND | ND |
| *Papaver nudicaule* | ND | ND | ND | ND | ND | ND | ND | ND |
| *Papaver pavonium* | ND | ND | ND | ND | ND | ND | ND | ND |

*The presence of the promorphinan and morphinan compounds in latex and capsule samples was determined by high resolution accurate mass spectrometry (HRAM) (Supplementary Data 1). + sign represents species where the compounds were quantitatively measured at a level 10x above the calculated limits of detection (LOD) and the ND (not detectable) represents species where the compounds were below our LOD. LOD values were calculated using authentic standards. Promorphinan compounds were identified in four species indicative of a functional *STORR*, with morphinan production identified in two species plus *P. somniferum* and *P. setigerum*. Source data are provided as a Source Data file.

harvest was carried out (Table 1 and Supplementary Data 1). As previously reported, we found thebaine to be the most prominent metabolite in *P. bracteatum* and oripavine the most prominent in *P. orientale*[19–21]. These two species along with *P. somniferum* and *P. setigerum*[14] were the only species identified as producing morphinans. A recent report of trace amounts of morphinans in *P. rhoeas* appears to have been conducted without the use of known morphinan standards[14]. Our analysis of *P. rhoeas* capsule material conducted using high resolution accurate mass LC-MS does not detect morphinans or promorphinans above defined limits of detection (Supplementary Data 1), which for morphine is 226-fold lower than morphine levels in *P. setigerum*. Our results of zero peak area for morphinans in *P. rhoeas* are in agreement with other published results[17,22,23] and consistent with the absence of morphinan-related genes in this species.

From the other species in our analysis, promorphinans were identified as minor compounds in *P. armeniacum* and *P. triniifolium* (Table 1 and Supplementary Data 1). Identification of these morphinan and promorphinan compounds is suggestive of the presence of a *STORR* ortholog which was selectively lost in some *Papaver* species.

Transcriptomic analysis revealed expression of *STORR* and either none, some or all of the promorphinan genes in a subset of the nine other *Papaver* species, with the full complement in opium poppy as reported previously[5]. These are shown on the branches of a species tree generated using conserved orthologs of 8 low-copy nuclear genes identified from the same transcriptomic datasets or the annotation of whole genome assemblies (Fig. 1b; Supplementary Tables 2 and 3, Supplementary Data 2 and 3).

The topology of the tree generated based on the species included is largely congruent with other phylogenetic trees constructed based on taxonomic sequence datasets of chloroplast, ribosomal, and plastid markers from the genus *Papaver* and the wider family and order[24–28]. The differences observed in the ordering of the *Papaver* species can be attributed to the marker sets and methods used for the assembly of the species tree and range of plant samples sequenced. The divergence times estimated between the species of the tree are in the regions of those previously estimated, for example the divergence of *P. californicum* at around 16.8 million years ago (MYA) compares to previously estimated timings[27–30].

**Papaver STORRs convert (S)- to (R)-reticuline in vitro.** Similar to the opium poppy *STORR*, the homologues present in the other five *Papaver* species (Fig. 1b) encode full length P450-oxidoreductase fusion proteins (Supplementary Fig. 1). The direction of both modules and the 9-13 amino acid linker sequences are also conserved (Fig. 2a; Supplementary Fig. 1) with overall sequence identity ranging from 89.4-99.9% at the nucleotide level and 86.0-99.9% with the predicted amino acid sequences (Supplementary Data 4).

The *STORR* gene fusion from *P. somniferum* and *P. bracteatum* have previously been shown to catalyse the gateway reaction to promorphinan biosynthesis[3,4,7]. The identification of *STORR* in conjunction with promorphinans in *P. armeniacum* and *P. orientale* suggest a similar role in these species. In *P. californicum*, *STORR* was expressed but promorphinans and genes related to promorphinans were not detected. However, the main BIA identified from our extraction of *P. californicum* was a member of the aporphine subclass, glaucine in the (R) configuration (Supplementary Table 1 and Supplementary Fig. 2).

In order to investigate the function of the *P. californicum* STORR, we expressed this gene in *Saccharomyces cerevisiae*, performed epimerization assays on microsomal fractions and found the same activity as the *P. somniferum*, *P. bracteatum* and *P. armeniacum* STORR proteins (Fig. 2b). Activity of *P. californicum* STORR microsomal fractions on alternative

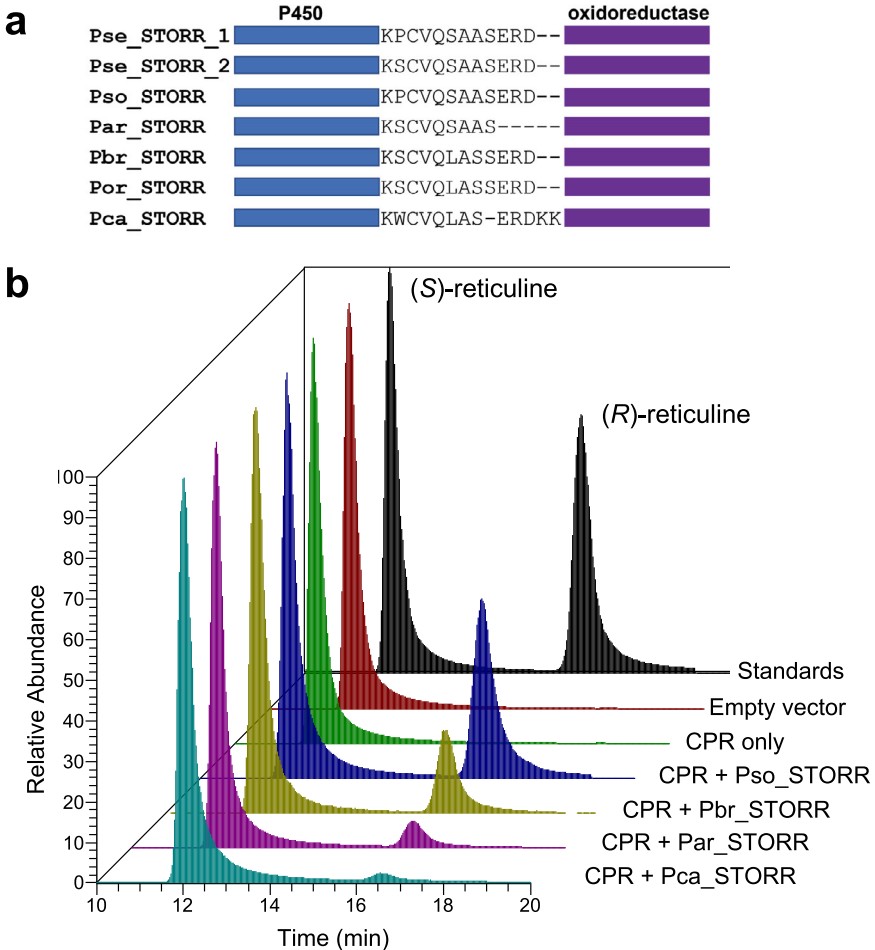

**Fig. 2 Functional analysis of STORR proteins in *Papaver* species. a** Alignment of the amino acid linker connecting the *STORR* P450 and oxidoreductase modules. Pse *P. setigerum*, Pso *P. somniferum*, Par *P. armeniacum*, Pbr *P. bracteatum*, Por *P. orientale* and Pca *P. californicum*. **b** Functional characterisation of the STORR proteins by heterologous expression in *S. cerevisiae*. STORR protein from *P. californicum, P. armeniacum, P. bracteatum* and *P. somniferum* were separately produced with a *P. somniferum* cytochrome P450 reductase (CPR) redox partner in *S. cerevisiae*. Microsomal proteins were assayed with (*S*)-reticuline as substrate. Relative abundance is used to show the reticuline epimers present in the sample. Reticuline standards (black), pESC-trp empty vector (red), pESC-trp vector + cytochrome P450 reductase (CPR)(green), pESC-TRP + CPR + Pso STORR (blue), pESC-TRP + CPR + Pbr STORR (gold), pESC-TRP + CPR + Pam STORR (pink) and pESC-TRP + CPR + Pca STORR (cyan). Source data are provided as a Source Data file.

substrates (*S*)-glaucine and (*S*)-laudanosine were also investigated with no activity found (Supplementary Fig. 3 and 4).

**Monophyletic origin of *STORR* in the genus *Papaver* 16.8 MYA.** To understand the evolutionary relationship, we performed gene tree analysis of the *STORR*s and the closest paralogues to the two *STORR* modules within the genomes of a representative subset of *Papaver* species. The presence of a closely linked gene pair of closest paralogs to the opium poppy *STORR* cytochrome P450 *CYP82Y2* and *oxidoreductase* modules was previously discovered through analysis of a whole-genome assembly[5]. The segmental duplication resulting in these paralogs was suggested to have occurred 20.0–27.8 MYA by the Ks estimation of the paralogous pairs[6]. In order to establish if equivalent paralogous pairs are present in related *Papaver* species we used whole-genome sequencing approaches to assemble draft genomes for *P. nudicaule* from Clade 1 and *P. californicum, P. bracteatum, P. atlanticum,* and *P. armeniacum* from Clade 2 (Supplementary Table 4). We compiled and annotated all homologous sequences that contain full-length genes corresponding to either of the *STORR* modules in these draft assemblies. We combined all *CYP82Y2* and *oxidoreductase* sequences

from the five draft genomes presented here with those retrieved from searches of the annotated opium poppy[5,14], *P. rhoeas* and *P. setigerum*[14] genomes, as well as the transcriptomic data of other species in the present study. We then constructed gene trees for the two gene subfamilies containing the coding sequences closely related to the *STORR* modules *CYP82Y2* (Fig. 3a) and *oxidoreductase* (Fig. 3b), respectively. Consistent with previous work[6], both trees revealed robustly supported cytochrome P450 *CYP82Y2-like* (*CYP82Y2-L*) and *oxidoreductase-like* (*COR-L*) groups. All P450 modules and oxidoreductase modules of *STORR*s are from Clade 2 *Papaver* species and they form monophyletic orthologous subgroups highlighted in orange (Fig. 3a, b).

The *STORR* subgroups contain no other complete coding sequences for P450 or oxidoreductase apart from Pca_CYP82Y2_Lstorr, which actually contains the conserved linker sequence and the first 13 codons of an oxidoreductase before a stop codon, indicating deletion after the fusion. Therefore these two monophyletic *STORR* subgroups and the presence of functional *STORR* in *P. californicum* (Fig. 2b) lead us to conclude that all *STORR* genes from these *Papaver* species have derived from a single fusion event in the common ancestor of Clade 2 species after their divergence from Clade 1 between 16.8 to 24.1

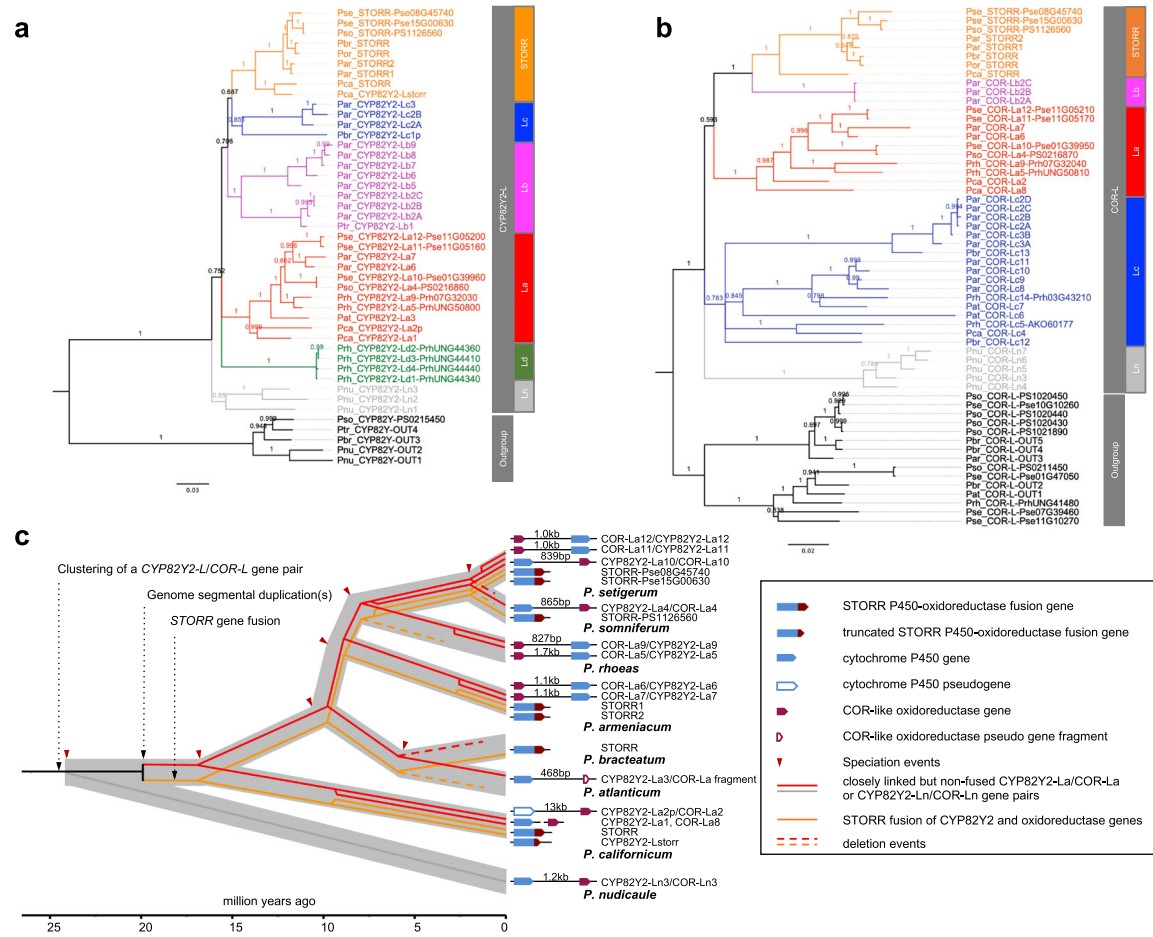

**Fig. 3 Evolutionary history of the formation of *STORR* fusion gene inferred from gene tree analyses of its P450 CYP82Y2 and oxidoreductase modules. a** Phylogenetic gene tree of *CYP82Y2-L* P450 subfamily constructed using Bayesian Inference. **b** Phylogenetic gene tree of *COR-L* oxidoreductase subfamily constructed using Bayesian Inference. The posterior probabilities are shown on the branches and scale bar represents substitutions per nucleotide site. Both trees were rooted based on branch position of orthologous outgroup containing *P. somniferum* sequences CYP82Y-PS0215450 for *CYP82Y2-L* tree and COR_L-PS0211450/PS1020430/PS1020440/PS1020450/PS1021890 for the *COR-L* tree as described in Li et al.[6]. *STORR*, La, Lb, Lc, Ld and Ln are defined as orthologous subgroups and their members are highlighted by coloured branches. A three letter prefix followed by an underscore is used as a species identifier for each gene; including Par *P. armeniacum*, Pat *P. atlanticum*, Pbr *P. bracteatum*, Pca *P. californicum*, Pnu *P. nudicaule*, Por *P. orientale*, Prh *P. rhoeas*, Ptr *P. triniifolium*, Pse *P. setigerum* and Pso *P. somniferum*. **c** Schematic representation of the evolutionary history of *STORR* reconciling the species tree with gene trees. The fusion event was preceded by the clustering of a gene pair of *CYP82Y2-L* and *COR-L* genes and subsequent segmental duplications. The grey background branches denote species divergence with speciation time points indicated by red arrows. The orange lines denote *STORR* and the red lines a gene pair of *CYP82Y2-La* and *COR-La*. The grey line indicates an ancestral *CYP82Y2-L* and *COR-L* gene pair prior to the divergence of *P. nudicaule* from the Clade 2 *Papaver* species. The exclusive presence of Clade 2 species in La and *STORR* subgroups and a single *P. nudicaule* Ln subgroup in both *CYP82Y2-L* and *COR-L* trees is consistent with the segmental duplication occurring after the divergence of *P. nudicaule* from Clade 2 species but before *STORR* formation between 24.1 and 16.8 MYA. Source data are provided as a Source Data file.

MYA (Fig. 3c). Our analysis of *STORR* gene evolution also reveals the importance of lineage-specific deletion (*P. atlanticum* and *P. rhoeas*), duplication (*P. californicum*, *P. armeniacum*, *P. somniferum* and *P. setigerum*) and rearrangement after duplication (*P. californicum*) (Fig. 3c). In addition, we note that in *P. somniferum* one copy of *STORR* has been lost after a whole-genome duplication event[5].

We also observed Clade 2 orthologous subgroups (*CYP82Y2_La* and *COR-La*, highlighted in red), which contains the closest paralogue pair (Pso_CYP82Y2-La4-PS0216860 and Pso_-COR_La4-PS0216870) from the opium poppy genome in the respective gene trees (Fig. 3a, b). This and the presence of the corresponding copies in *P. californicum* indicates all sequences within the subgroups are derived from a single copy Clade 2 ancestor as found to be the case for *STORR*. We found the same pairing arrangement of closest paralogue pairs in all our assembled genomes except *P. bracteatum* which appears to have lost this paralogous gene pair (Fig. 3c). In *P. californicum* and *P. atlanticum* we found evidence for the ancestral arrangement of the paralogous gene pairs but in both cases one member of the pair now is present as a pseudogene (Fig. 3c). The order of the P450 and oxidoreductase genes for both paralogous pairs in *P. armeniacum* and *P. rhoeas*, as well as two of the three gene pairs in *P. setigerum*, have switched compared to the other species (Fig. 3c). Taken into consideration gene deletion, duplication, rearrangement and erosion it is still apparent from these findings that paralogous pairing of a *CYP82Y2_La* and *COR_La* existed in the Clade 2 common ancestor (Fig. 3c), supporting the hypothesis that a segmental duplication would have occurred prior to *STORR* fusion/neofunctionalization.

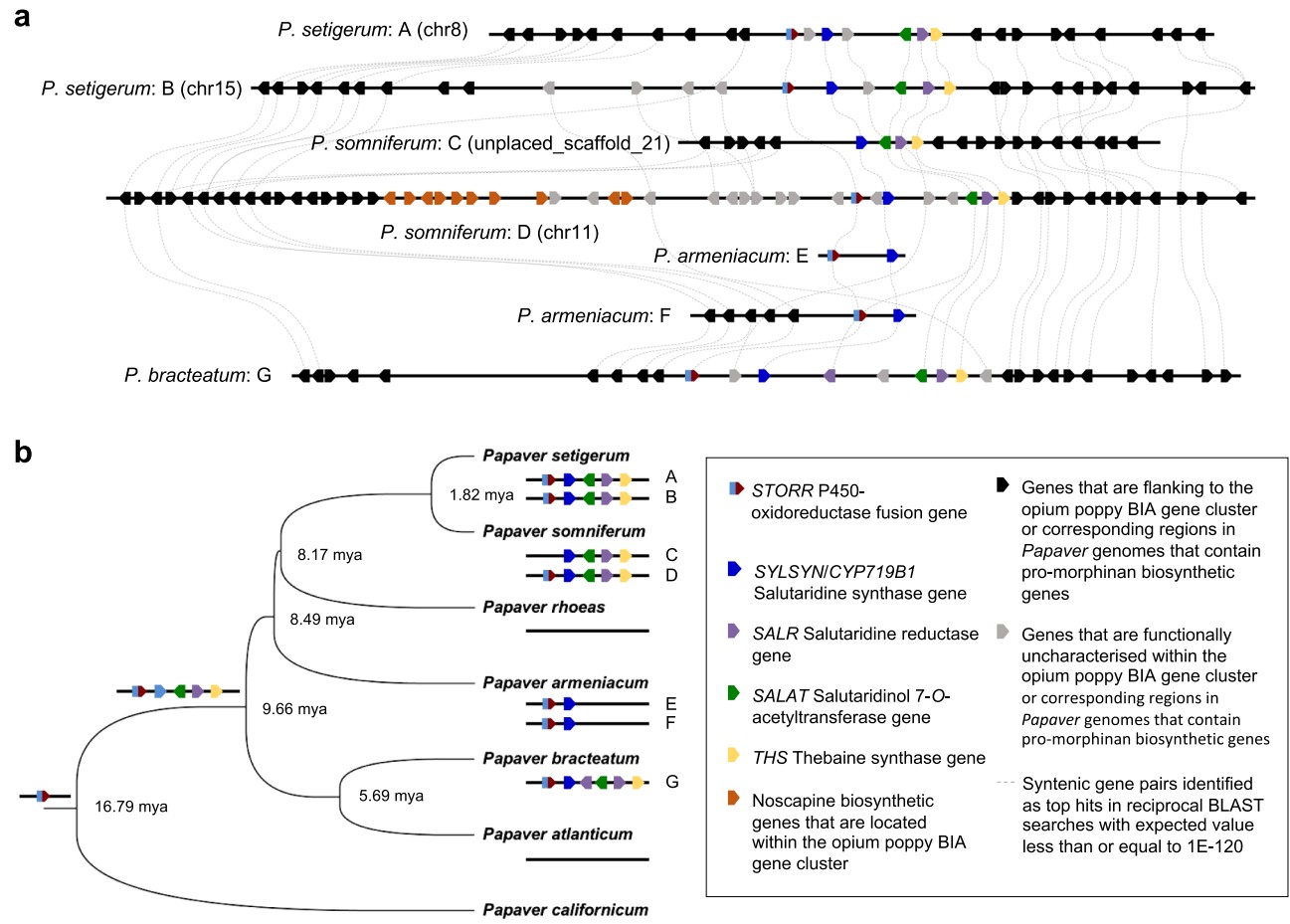

**Fig. 4 Evolutionary history of the STORR plus promorphinan component of the BIA gene cluster in *Papaver*. a** Graphic representation of synteny between contigs that contain *STORR* and promorphinan genes in *Papaver* genomes. Arrangement of the genes and their orientations of the syntenic regions of *Papaver* species are shown as indicated in the inset (Supplementary Data 5); including the corresponding regions in the opium poppy genome, one of which contains the BIA gene cluster and its flanking regions (*P. somniferum*: C&D)[5], the two syntenic regions in the *P. setigerum* genome on Chromosomes 8 and 15 (*P. setigerum*: A&B)[14]; the two contigs of 94 kb and 294 kb from the *P. armeniacum* genome (*P. armeniacum*: E&F) and the single 1.2 Mb contig of the *P. bracteatum* assembly (*P. bracteatum*: G). Dashed lines denote syntenic gene pairs. **b** Summary of the evolutionary history of the *STORR* plus promorphinan genes of the BIA gene cluster in *Papaver* as inferred from the species phylogeny and synteny in the genomes. The phylogeny for the selected subset of *Papaver* species and divergent times at the branching points were extracted from the species tree in Fig. 1b. The emergence of *STORR* and its clustering with all four promorphinan genes are indicated on the ancestral nodes. These are derived from the presence and organisation of these genes in the genomes of these extant species, which are also shown. Source data are provided as a Source Data file.

Within both *CYP82Y2-L* and *COR-L* groups, the gene trees have identified single orthologous groups (*CYP82Y2-Ln* and *COR-Ln*, highlighted in grey) containing exclusive Clade 1 *P. nudicaule* sequences, which are absent from all other subgroups (Fig. 3a, b). Among the members, there is at least one paralogue pair (*Pnu_CYP82Y2_Ln3/Pnu_COR_Ln3*) just over 1 kb apart (Fig. 3c), which was confirmed by direct sequencing of a PCR amplified genome fragment. This finding implies that the origin of the *CYP82Y2_L* and *COR_L* pairing event occurred before the separation of *P. nudicaule* from Clade 2 *Papaver* species 24.1 MYA. This is consistent with the presence of additional paralogous pairs identified in the Clade 2 Lb and Lc subgroups which will have arisen by multiple segmental duplication events (Fig. 3a, b). One such duplication event will have led to the formation of *STORR* 16.8–24.1 MYA.

**STORR clustered with four promorphinan genes 9.7 MYA.** As part of the 800 kb BIA cluster, *STORR* is clustered with four promorphinan genes in the opium poppy genome, which also

contains a second syntenic region containing paralogs of the promorphinan genes but not *STORR*[5] (Fig. 4a). Both these syntenic regions are well conserved in *P. setigerum*, the closest related sister species of opium poppy, with one on chromosome 15 and the other on chromosome 8 (Fig. 4a; Supplementary Data 5). Both syntenic regions in *P. setigerum* contain *STORR* and all four promorphinan genes. Two annotated genes, Pse08G45690.0 and Pse08G45700.0, showing similarity to *SALAT* are arrayed in tandem in the syntenic region on *P. setigerum* Chromosome 8 (Supplementary Data 5). Close inspection using sequence alignment indicates that both Pse08G45690.0 and Pse08G45700.0 are mis-annotated in the reported annotation[14] and instead represent two fragments of a continuous intronless *SALAT* pseudogene, containing one frameshift and one nonsense codon mutation in the open reading frame. Nucleotide sequence alignment of *STORR* genomic DNA regions suggests that the *STORR* copy Pse08G45740.0 in the *P. setigerum* chromosome 8 cluster is also likely to be a pseudogene as it contains an additional insertion in the open reading frame besides the introns at the conserved positions. We found that *P. bracteatum* shows good synteny with

the opium poppy regions containing the promorphinan genes with a notable difference being an extra copy of *SALR* in the former (Fig. 4a; Supplementary Data 5). *P. armeniacum* shows synteny with *STORR* and *SALSYN* with gene order conserved in two smaller contigs, one of which extends to reveal five additional genes in the flanking region. Only *STORR* was found in the draft assembly of *P. californicum*, the *P. atlanticum* draft assembly lacks *STORR* and the promorphinan genes (Supplementary Data 2). These findings are consistent with our metabolite profiling and transcriptomic analyses across these five species (Table 1; Fig. 1b), highlighting the role played by gene deletion events in formation of the diversity of BIA composition within the *Papaver* lineage.

Together these findings lead us to conclude that the *STORR* plus promorphinan component of the BIA gene cluster formed prior to the divergence of *P. bracteatum* and *P. somniferum* 9.7 MYA (Fig. 4b), but after the formation of *STORR* at least 16.8 MYA. Orthologues of *SALSYN*, *SALAT*, *SALR* and *THS* have only been identified in the genomes of a subset of the Clade 2 *Papaver* species that produce promorphinan/morphinan compounds and contain the *STORR* gene fusion (Supplementary Data 2). Percentage nucleotide and amino acid identity matrix of these orthologues range from 94.1-99.8% and 90.5-100% respectively (Supplementary Data 4), supporting the case for evolution of the promorphinan/morphinan biosynthetic pathway being triggered after the *STORR* gene fusion event.

## Discussion

We demonstrate a functional *STORR* protein in *P. californicum*, which has persisted since its divergence from the common ancestor 16.8 MYA. *P. californicum* represents the earliest branching species in the *Papaver* lineage that we have found to contain the *STORR* gene fusion event occurring between 24.1 − 16.8 MYA (Fig. 1b). While we are confident that the *STORR* gene fusion occurred before the neofunctionalization of promorphinan genes we cannot conclude whether the latter was before or after the branching of *P. californicum* from the other Clade 2 species as shown in Fig. 1. Therefore it is possible that either neofunctionalization did not happen prior to the branching of *P. californicum* or it did happen and the promorphinan genes were lost.

Genome rearrangements result in considerable structural variation even within a single species as evidenced by the loss of the noscapine component of the BIA gene cluster[9] and significant variation in copy number of genes associated with morphinan biosynthesis in *P. somniferum*[6]. That *STORR* persists in *P. californicum* suggests it is providing some selective advantage.

*P. californicum* is a *Papaver* species indigenous to California ('New World') thought to have evolved in parallel to the Eurasian 'Old World' members of the family[24,27,29,30]. The specific distribution of *P. californicum* to North Western America is an example of an 'Old World/New World' disjunction at 28 − 10 MYA in the Papaveraceae distribution[30]. The occurrence of such North American/Eurasian disjunctions is recognised in a number of species, and attributed to historic changes in climate and existence of previous land-bridge connections[30–33]. The differences observed in the metabolite profile of *P. californicum* compared to the morphinan-producing species could be attributed to its parallel evolution in a different environment with different selective pressures.

We found the most abundant BIA in *P. californicum* to be (*R*)-glaucine, which to our knowledge has not previously been reported in nature. Glaucine in the (*S*) configuration isolated from *Glaucium* species in the Papaveraceae is associated with bronchodilator, anti-inflammatory and neuroleptic effects[34,35]. A chemically synthesised form of (*R*)-glaucine has been shown to increase the efficacy of serotonin[36]. Evidence for (*S*)-glaucine biosynthesis via (*S*)-reticuline has previously been proposed[37]. It is therefore interesting to speculate, based on the emergence of *STORR* at 16.8 MYA in the *Papaver* lineage, that the *P. californicum STORR* could be involved in (*R*)-glaucine formation. Epimerization assays using *P. californicum STORR* with (*S*)-glaucine and (*S*)-laudanosine as substrate showed no activity implying that neither of these are intermediates in the biosynthesis of (*R*)-glaucine but rather the pathway is dependent on epimerization of reticuline by *STORR* followed by formation of the tetracylic aporphine structure.

A recent report based on genome comparison of opium poppy with two very closely related *Papaver* species, *P. setigerum* and *P. rhoeas* has proposed the fusion event that resulted in *STORR* occurred following the whole-genome duplication event in opium poppy after its divergence from *P. rhoeas*[14]. However, that proposal does not consider the possibility of *STORR* gene loss within *P. rhoeas* nor does it take into account the previously reported presence of *STORR* in *P. bracteatum*[7] which diverged earlier from all three species (Fig. 1b)[24]. Furthermore, that proposal associates the segmental duplication giving rise to the *CYP82Y2-La/COR-La* gene pair and *STORR* with the opium poppy WGD event but no supporting evidence was presented[14]. Figure 3 demonstrates that the required gene duplication event must have happened before this WGD event.

Our findings with wider taxonomic sampling clearly show that the *STORR* gene fusion was a single event that occurred between 16.8-24.1 MYA in the *Papaver* lineage and was preceded by clustering and segmental duplication of the P450 oxidase and oxidoreductase genes that fused to form it. While the *STORR* gene fusion is regarded as a key event enabling the evolution of the morphinan subclass of BIAs our findings show that it may also have enabled the production of compounds in other BIA subclasses such as the aporphine (*R*)-glaucine in *P. californicum*, which 16.8 MYA branched from the common ancestor of those other *Papaver* species that now produce morphinans.

## Methods

**Plant material.** The plant material used in the current study were voucher specimens sourced primarily from botanic institutions as detailed in Supplementary Table 1. Plants were grown under controlled long day conditions in the glasshouse facilities and in the experimental gardens located in the University of York. Samples for DNA cDNA and RNA were collected from young leaves of juvenile plants and flash frozen in liquid nitrogen. Metabolite analysis was conducted on latex harvested from three individual glasshouse grown plants at 70 days post germination and from capsules pre-dehiscence.

**Genomic DNA and cDNA isolation for PCR.** Genomic DNA was extracted from frozen ground young leaf using the BioSprint 15 Plant Kit on the BioSprint 15 Workstation (Qiagen, Crawley, UK). The DNA was quantified on the nanodrop 1000 (Thermo Fisher). cDNA was synthesised from RNA isolated from young leaf tissue using superscript V (Thermo Fisher) and used for gene-specific amplification.

**Metabolite profiling.** Metabolite extraction and UPLC analysis were carried out on both latex and dried capsule material. Fresh latex samples were collected from the stems of juvenile plants into 10% acetic acid for extraction. Following flowering capsule material was also collected, dried and ground. A 10 mg sample of material from the dried capsules was then extracted in 10% acetic acid. The extracts were analysed using an Acquity UPLC system (Waters Ltd., Elstree, UK) linked to a Thermo LTQ Orbitrap (Thermo Fisher, Hemel Hempstead, UK). Authentic standards for salutaridine and salutaridinol (Toronto research chemicals, Canada), and also a MCONT standard mix (morphine, codeine, oripavine noscapine and thebaine) were included to confirm the presence or absence of promorphinan and morphinan compounds in the 10 *Papaver* species. Limit of detection for all standards were calculated by preparing each individual standard to a top concentration of 0.05 mg/ml in 10% Acetic acid equating to 100 ng on column in a 2 µL injection. From each of the top standards a 20 point serial dilution was prepared. The limit of detection (LOD) calculation was based on the standard deviation of the response (Sy) of the curve and the slope of the calibration curve (S) at a level approximating the LOD (according to the formula: $LOD = 3.3(Sy/S)$ Supplementary Data 1)[38].

**Isolation of glaucine by preparative HPLC**. A total of 20 g of dried capsule material from *P. californicum* or *G. flavum* plants was ground to a fine powder and extracted with 25mls of 10% acetic acid in water. The plant material was removed by centrifugation at 4000 g for 10 minutes and the extract dried down using an EZ-2 elite genevac. The dried residue was taken up in 2 ml of ethyl acetate and further spun to remove debris prior to purification. Isolation and purification of glaucine was performed using the interchim puriflash 4500 prep HPLC system with Advion Expression, compact mass spec (CMS). The ethyl acetate extracts were applied to a 12 g BUCHI FlashPure EcoFlex silica 50 μm irregular column and all fractions collected using a 0-100% ethyl acetate in hexane gradient, followed by isocratic 100% ethyl acetate and 100% methanol. Samples from fractions thought to contain the glaucine peak (356 ion) were confirmed by running on an Acquity UPLC system (Waters, Elstree, UK) linked to a Thermo LTQ Orbitrap (Thermo Fisher, Hemel Hempstead, UK) on an Acquity BEH c18 1.7 μm 2.1 × 100mm column with the mass spec using APCI ionization in positive polarity. The glaucine containing fractions were dried down and resuspended in 10% acetic acid for further purification on a 4 g TELOS Flash C18 column with fractions collected using a 2-80% gradient of Solvent B in Solvent A (where Solvent A is 10 mM Ammonium bicarbonate pH10.2; Solvent B is 100% methanol). This method yielded 3 mg of glaucine from *P. californicum* and 2 mgs from *G. flavum*. These were resuspended in 100% ethanol to a 2mg/ml final concentration for circular dichroism (CD) analysis.

**Analysis of glaucine extracts by circular dichroism**. CD spectra were collected on a Jasco J-1500 Circular Dichroism Spectrometer. The system was purged with oxygen-free nitrogen before lamp ignition and the lamp allowed to stabilise for at least 10 minutes before acquisition. A nitrogen purge flow of at least 5 L/min was employed during data collection. Temperature was regulated by a Peltier thermostat monitored at the cell holder at 20 °C.

CD spectra were collected from 400 nm down to 190 nm which was the limit of solvent absorption allowing a HT maximum of 600 V; scan speed was 50 nm/min, bandwidth 1 nm and 5 scans were averaged. Samples were measured in 1 mm path length quartz cuvettes at the specified concentration. Blank spectra were collected in the same cuvette as the sample using the same solvent, and subtracted from the sample spectra. Data analysis was performed with Jasco Spectra Manager v2 software. UV absorption spectra of the samples were collected on a Jasco V560 spectrophotometer in the same cuvettes over the range 400 to 210 nm.

**RNA-Seq sequencing and transcriptomic data analyses**. Transcriptomic analysis was performed on RNA extracted using the Direct-zol RNA Miniprep Kit (Zymo Research, USA) according to the manufacturer's instructions from young frozen leaf material. RNA was quantified on the Qubit 3.0 Fluorometer (Thermo Fisher Scientific) according to the manufacturer's protocol and quality assessed by running 1 μL on the agilent tapestation.

RNA sequencing was performed on the nine *Papaver* species (Supplementary Table 1). RNA-Seq libraries were prepared from 1 μg high-quality RNA using the NEBNext Poly(A) mRNA Magnetic Isolation Module and NEBNext Ultra II Directional RNA Library Prep Kit for Illumina (New England Biolabs), according to the manufacturer's guidelines. Libraries were subject to 150 base paired end sequencing on one lane of a HiSeq 3000 system at the University of Leeds Next Generation Sequencing Facility (Leeds, UK) except for *P. californicum* which was sequenced on NovaSeq 6000 at Novogene Co, Cambridge, UK. An average of 7.5 Gb PE reads sequencing data were generated for each species (Supplementary Table 2).

Transcriptomes were assembled with the Trinity (v2.2.0) RNA-Seq De novo Assembly software pipeline[39] after filtering out any of the remaining 1–5% ribosomal RNA in the raw reads for each species with mapping to rRNA_115_tax_silva_v1.0 downloaded from SILVA database (https://www.arb-silva.de/). These transcriptome datasets were used for identification of orthologous genes of selected BIA biosynthetic genes (Supplementary Table 3) and the eight conserved ortholog sets (COS) genes[40,41] (Supplementary Data 3) by local BLAST searches. Sequences of the top matches were retrieved. Orthologous gene sequences belonging to gene families were identified and verified through subsequent gene tree analysis along with the related gene family's datasets that were previously reported[6] (Supplementary Data 2).

**Species phylogeny and estimation of divergence times**. Estimations of divergence dates among the nine *Papaver* species together with *P. rhoeas*, *P. setigerum* and opium poppy were based on the 8 COS gene sequences from the transcriptomic datasets (Supplementary Data 3) along with other Papaveraceae species *E. californica*, *M. cordata* and one Ranunculaceae species *A. coerulea* with BEAST2 v2.5.1[42].

Orthologous sequences of the eight COS genes were identified and retrieved from the annotated gene datasets of *P. somniferum*, *E. californica*, *M. cordata* and *A. coerulea* after conducting BLAST searches. Multiple sequence alignments of each gene set were obtained with MUSCLE v3.2[43,44]. Conservative alignment blocks were generated with Gblocks v0.91[45] to remove highly polymorphic regions and the alignments of all eight genes were subsequently concatenated in MegaX[46].

Species divergence times were estimated under the strict clock model implemented in BEAST v2.5.1 to generate a Bayesian posterior sample of time-calibrated phylogenies and the associated maximum clade credibility tree using two prior calibration points (Ranunculales 110 ± 5 MYA and Papaveraceae 77 ± 4 MYA[6]). Priors were treated as fitting a Yule speciation process and lognormal distribution. The nucleotide substitution model used was GTR with 4 Gamma categories. The chain length of Markov chain Monte Carlo (MCMC) was set to 30,000,000 runs, performed per 10,000 generations and collected every 1000th generation; with 10% of the total trees discarded as burn-in samples, the remaining trees were used for generating the consensus tree. Convergence of the runs performed by BEAST was assessed by effective sample sizes (ESS) using Tracer v1.5[42]: ESS exceeded 1000 for all summary statistics, greatly above the threshold of 200 that is considered to indicate good sample quality. The species tree and divergence times were visualised using Figtree v1.4.3[47].

**Heterologous expression of *Papaver STORR* for functional analysis**. Full-length cDNAs for *STORR* were cloned and sequenced after PCR amplification from *P. californicum*, *P. armeniacum* and *P. bracteatum* using the following primers (Integrated DNA Technologies - standard desalted): Pbra_*STORR* Fwd, 5'-ATGG AGCTCCAATATTTTTC-3'; Pbra_*STORR* Rev, 5'-TCAAGCTTTGTCATCCC A-3'; Pcal_*STORR* Fwd, 5'-ATGGAGCTCCAGTACTTTTC-3'; Pcal_*STORR* Rev, 5'-TCAAGCTTTGTCATCCCAC-3'; Parm_*STORR* Fwd, 5'-ATGGAGCTCCAAT ATTTTTC-3'; and Parm_STORR Rev, 5'-TTCAAGCTTTGTCAT-3'.

Percentage identity matrix between pairs of *STORR* homologues in *Papaver* species were calculated with pairwise sequence comparison tool EMBOSS Matcher version 2.0u4[36] (Supplementary Data 4). Multiple sequence alignments of the amino acid sequences were obtained with MUSCLE v3.2[43,44].

Geneart gene synthesis service (Thermo Fisher) created synthetic DNA sequences for the *STORR* gene of *P. californicum*, *P. armeniacum* and *P. bracteatum* identified from transcriptome sequence data. The sequences were codon optimised for expression in *Saccharomyces cerevisiae*. The codon optimised sequences are provided in Supplementary Data 6. The codon optimised genes were cloned from the Geneart pMK vector by digesting with NotI and PacI and inserting the cleaned DNA fragments behind the Gal10 promoter of the pESC-TRP expression vector. Vectors were sequenced for errors before being transformed into the *S. cerevisiae* G175 using lithium acetate protocol[48]. Yeast cultures were grown, and microsomal preparations were performed; the resulting crude microsomal preparations (12 mg mL-1 protein) were used for enzyme assays using 100um (*S*)-reticuline for the chiral analysis of the reticuline epimers. Reticuline was extracted from the assay reactions with dichloromethane, and epimers analysed by chiral LCMS[3], with the following modifications: a Lux 250 × 4.6 mm 5 u Cellulose-3 column (Phenomenex) was used, with LC performed using a Waters Acquity I-Class UPLC system interfaced to a Thermo Orbitrap Fusion Tribrid mass spectrometer under Xcalibur 4.0 control. MS1 and data dependent MS2 spectra were collected at 60000 resolution (FWHM), with the precursor m/z 330.1700 added to the acquisition method inclusion list. (*S*)-reticuline eluted at ~12 min and (*R*)-reticuline at ~16 min. Data was processed using Xcalibur software.

**Whole genome sequencing and assemblies**. 10X Genomics whole genome sequencing and assemblies were performed on five *Papaver* species, including *P. nudicaule*, *P. californicum*, *P. atlanticum*, *P. bracteatum* and *P. armeniacum*. Further Oxford Nanopore Technology (ONT) sequencing and genome assemblies were carried out on two of these species *P. bracteatum* and *P. armeniacum*.

High molecular weight (HMW) genomic DNA was prepared for 10X Genomics sequencing for five species. Young seedling material was grown and sent to Amplicon Express (Pullman, WA, USA), where HMW genomic DNA was prepared by using their proprietary protocol for HMW grade (megabase size) DNA preparation. This protocol involves isolation of plant nuclei and yields pure HMW DNA with >100 kb fragment length. The DNA samples were then used to construct the 10X Chromium libraries, which were subsequently sequenced on Illumina NovaSeq platform to produce 2 × 150 bp reads at HudsonAlpha Institute for Biotechnology, Huntsville, Alabama. This produced a total of over 2 billion x2 reads or 600 Gb of 10X Chromium library sequencing data for each species.

We used GenomeScope v1.0[49] to estimate the genome size and heterozygosity level for each species. Firstly, a k-mer distribution was generated with Jellyfish[50] from the 10X Genomics short reads, and then used as input in the subsequent GenomeScope analysis (Supplementary Table 4). De novo draft 10X assemblies were produced with the linked reads at approximately 50-60 times coverage of the estimated genome size, as required by the Supernova assembly software package (Supplementary Table 4)[51] for optimal performance.

Further sequencing of *P. bracteatum* and *P. armeniacum* was carried out using ONT long read sequencing platform by the Technology Facility at the University of York. HMW DNA samples were prepared from frozen young leaf tissue using a CTAB extraction method followed by a Qiagen genomic tip clean-up and removal of short DNA fragments using the Circulomics SRE kits (PacBio)[52]. Long read sequencing was performed by the University of York Bioscience Technology Facility Genomics lab, using the Oxford Nanopore Technologies MinION and PromethION platforms. Sequencing libraries were prepared using ONT's ligation sequencing kit SQK-LSK106, using a minimum of 5 μg high-quality DNA. The resulting library was split into 4 to allow loading onto MinION (FLO-MIN106) and

PromethION (FLO_PRO002) R9.4.1. flow cells, and sequencing for 48 hours (MinION) or 72 hours (PromethION), with a flow cell wash (using ONT wash kit EXP-WSH003) and library reloading step approximately 24 hours into the runs. Basecalling was performed using ONT's Guppy toolkit, version 4.0.11. Just over 7 million reads were generated with average read length of 13,395 bp and 95 Gb in total for *P. bracteatum*, whereas 4.5 million reads with average length of 22,216 bp covering 99 G nucleotide bases for *P. armeniacum*.

We ran the FLYE version 2.8-b1674[53] de novo assembly pipeline with the ONT datasets, producing initial assemblies. This was followed by one round base polishing with the starting ONT raw reads using RACON (https://github.com/isovic/racon). Purge_dups[54] was then used to remove haplotigs and contig overlaps to produce a haploid representation of the genome. Final draft genome assemblies (Supplementary Table 4) were achieved after two more rounds of base polishing using FREEBAYES software tool[55] after the Illumina short reads from the 10X Chromium libraries were mapped to the working assembly with Longranger software package (https://support.10xgenomics.com/genome-exome/software/pipelines/latest/what-is-long-ranger).

The presence of orthologous genes of selected BIA biosynthetic genes (Supplementary Table 3) were conducted firstly by local BLAST searches in the 10X assemblies of *P. nudicaule*, *P. californicum*, and *P. atlanticum*, the final draft ONT assemblies of *P. bracteatum* and *P. armeniacum* and the annotated gene sets of *P. rhoeas* and *P. setigerum*[14]. Sequences of the top matches were retrieved and verified through subsequent gene tree analysis along with the related gene families datasets that were previously reported[6] (Supplementary Data 2).

**Repeat annotation and gene prediction**. For each of the five assembled *Papaver* genomes, a repeats library was constructed ab initio using RepeatModeler (v2.0.1, http://www.repeatmasker.org/RepeatModeler)[56]. The consensus TE sequences generated by the RepeatModeler software were then used as repeats library in RepeatMasker (v4.1.1, http://www.repeatmasker.org) to identify repetitive elements in all five genomes. In addition, we identified intact LTR-RTs using LTR_retriever (v2.8)[57], which integrates results of LTR_FINDER (v1.1)[58] and LTR_harvest (v1.5.9)[59]. Insertion times of LTRs were also calculated by LTR_retriever with parameter -u set to 1.396e-8. An integrative approach combining homology-based, RNA-seq-based, and ab initio gene prediction was used to identify and predict protein-coding genes in the *Papaver* genomes. Genome sequences and gff files of *Arabidopsis thaliana* and two previously published *Papaver* genomes, *P. somniferum* and *P. rhoeas*, were used for homology-based prediction using GeMoMa[60] (v1.8, http://www.jstacs.de/index.php/GeMoMa) with default parameters. Illumina RNA-seq reads from young frozen leaf material that were used for transcriptome assemblies were mapped to the genome using HISAT2[61] (v2.1.0) and the alignments were assembled with StringTie2[62] (v2.1.5). Transdecoder (v5.0.2, https://github.com/TransDecoder) was then used to find potential open reading frames (ORFs). For ab initio predictions, BRAKER2[63] was used and model training was based on RNA-seq data after the predicted repeats were soft-masked within the genome assembly. Finally, all structural gene annotations were joined with EvidenceModeler[64] (v1.1.1), and weights were assigned as follows: RNA-seq: 5; homology-based: 5; ab initio: 2. Functional domains of protein-coding genes were identified with InterProScan version 5.52-86 with default parameters. BUSCO analysis was run on the finalised annotations of the five *Papaver* genomes using BUSCO version 4.1.4 and embryophyta_odb10 used to estimate genome completion[65].

**Gene tree analyses of *CYP82Y2-L* and *COR-L* gene subfamilies**. Gene tree analyses were used to understand the evolutionary events leading to *STORR*s in *Papaver* species and to reconstruct a detailed evolutionary history by analysing the entire set of homologous representatives to *STORR*s and their closest paralogous *CYP82Y2-L&COR-L* gene pairs using a total of eight representative *Papaver* genomes, by reconciling the gene duplication events implied in gene trees with the *Papaver* speciation tree with the parsimony principles.

We searched the above five draft genome assemblies with BLASTN to identify regions corresponding to either of the *STORR* modules. The sequences of these regions were then retrieved and manually annotated with their sequence homology to other closely related genes. All *CYP82Y2-L* and *COR-L* homologous sequences that contain full length coding regions were verified in annotated gene datasets and retained for the further gene tree analysis.

As the draft *P.nudicaule* 10X assembly is highly fragmented, likely caused by high heterozygosity level in its genome, we obtained full length coding regions for all potential candidates by PCR amplification from both cDNA libraries and genomic DNA using gene specific primers. Fragments to confirm the *P. nudicaule* CYP82Y2 and cor modules was carried out by RT-PCR and sequencing of fragments using the following primers (Integrated DNA Technologies - standard desalted): PN_CYP + COR_1717256_Fwd, 5'-ATGGATTACTCTAATCTTCAG-3' and PN_CYP + COR_1717256_Rev, 5'-GTCATCCCACACCTCTTC-3'. The amplified fragments were cloned and sequenced for verification.

We also retrieved *CYP82Y2-L* and *COR-L* homologous sequences after BLASTN searches of the annotated *P. rhoeas*, *P. setigerum* and opium poppy genomes and the transcriptomic data from the other *Papaver* species in the present study. Reiterative gene tree analyses were carried out to identify the orthologous *CYP82Y2-L* and *COR-L* sequence subsets. Duplicated and partial sequences were

removed. These were combined with the above set from the five *Papaver* genomes (Fig. 3a, b) for the final gene tree analyses.

Multiple sequence alignments of the coding sequence sets of the *CYP82Y2-L* and *COR-L* were built with MUSCLE v3.2[43,44] and conservative alignment blocks were generated with Gblocks v0.91[45] by removing highly polymorphic regions. The gene trees were constructed using Bayesian analyses with MrBayes v3.2.6[66]. The trees were then visualised using Figtree v1.4.3[47]. Posterior probabilities were reported as supporting values for nodes in the trees and scale bar represents substitutions per nucleotide site (Fig. 3a, b).

The sequence regions containing all paralogous *CYP82Y2-L&COR-L* gene pairs were retrieved from the draft genome assemblies and manually annotated (Fig. 3c).

**Synteny analysis of *STORR* plus promorphinan component of the BIA gene cluster among *Papaver* genomes**. To investigate the synteny in the genomic regions containing *STORR* or the four promorphinan biosynthetic genes, we compared the gene order and orientation in the opium poppy 800 kb BIA cluster and flanking regions to the contigs/scaffolds in the five representative *Papaver* genomes. The contigs/scaffolds were identified after BLAST searches for containing *STORR* or *SALSYN/SALAT/SALR/THS* genes in the draft genome assemblies and their sequences were retrieved (Supplementary Data 4). A single contig was identified in *P.bracteatum* genome and its length is 1.2 Mb, whereas the single scaffold in *P.californicum* is the shortest with 44 kb in length. Two contigs were found in *P.armeniacum* to contain *STORR/SALSYN* genes, one is 294 kb long and the other 94 kb in length.

Ab initio gene prediction of these contigs and scaffolds was carried out using FGENESH, a web-based gene annotation tool[67] (http://www.softberry.com/berry.phtml?topic=fgenesh&group=programs&subgroup=gfind) with Dicot plants option as training set. The predicted genes were functionally annotated by homologous BLAST searches in the swissprot database. Reciprocal BLASTN searches were performed between the coding sequence sets of predicted genes in these contigs/scaffolds and the annotated opium poppy genome[5] (http://bigd.big.ac.cn/search?dbId=gwh&q=opium%20poppy&page=1). Similar reciprocal BLASTN searches were also performed between the pair of annotated datasets of *P. rhoeas*/*P. somniferum* and *P. setigerum*/*P. somniferum* genomes. The top match records in both sets with an expected value less than 1E-120 corresponding to the opium poppy BIA gene cluster region were summarised in Supplementary Data 5 and Fig. 4a.

**Reporting summary**. Further information on research design is available in the Nature Research Reporting Summary linked to this article.

## Data availability
The transcript and genome assembly data generated in this study have been deposited in the NCBI databases under the BioProject PRJNA770669. Raw RNA-seq and DNA reads are available under the accession numbers SRR16389878–SRR16389886, SRR16690173, SRR16690174 and SRR16591802–SRR16591806. Assembled RNA-seq datasets are available under the following accession numbers GJOO00000000–GJOY00000000. Annotated genomes are available under the following accessions (JAJJMA000000000, JAJJMB000000000, JAJJMC000000000, JAJJWW000000000, and JAJJWX000000000. Annotated genomes have also been made available at ORCAE[68] (Online Resource for Community Annotation of Eukaryotes) [https://bioinformatics.psb.ugent.be/gdb/poppy/]. Individual sequences of *STORR* gene have been deposited at NCBI under accessions OK631703–OK631705. In addition, individual sequences of *Papaver nudicaule COR-L* gene have been deposited at NCBI under accessions OK631706–OK631710, OK999969–OK999970, and OL452058. Source data are provided with this paper.

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

## Acknowledgements

I.A.G. received support from the Biotechnology and Biological Sciences Research Council, United Kingdom (grant BB/K018809/1) and the Garfield Weston Foundation, United Kingdom. We are grateful to York Technology Facility, COEMS for support with metabolite profiling and RNA-seq library preparation, the University of York High Performance Computing service, Viking and the Research Computing team for support with bioinformatics and J. Mitchell for administrative support. Y.V.d.P. acknowledges funding from the European Research Council (ERC) under the European Union's Horizon 2020 research and innovation programme (No. 833522) and from Ghent University (Methusalem funding, BOF.MET.2021.0005.01).

## Author contributions

T.C. and F.M. conducted all RNA and genomic DNA preparation, molecular biology, biochemical and enzyme analysis, T.C., F.M., D.H. and T.R.L. conducted all metabolite analysis, T.C. and Y.L. all transcriptomic analysis, Y.L. and A.C. species phylogeny analyses, Y.L. all gene tree analyses, Y.L. and Z.N. all draft genome assemblies and Y.L. all synteny analysis. J.C. and Y.V.d.P. performed all genome annotations. A.L. conducted circular dichroism analysis of glaucine extracts. T.C., Y.L., T.W. and I.A.G. analysed and interpreted results and were major contributors in writing the manuscript. All authors read and approved the final manuscript.

## Competing interests

The authors declare no competing interests.
