## [Peer Review File · Nature Communications]

A functionally conserved STORR gene fusion in Papaver species that diverged 16.8 million years agoReviewers' Comments:

Reviewer #1:

Remarks to the Author:

This is an interesting study on the evolution of the STORR gene in the genus *Papaver*.

In general, the ms is well written and the methods used for analysis are adequate.

The following minor issues need attention.

1. make clear what you consider a morphinan alkaloid
2. Only morphine is a strong analgesic within morphinans. It is only produced by *P. somniferum*. The sentence in the ms is misleading.
3. if you discuss the phylogeny of *Papaver*, you might discuss previous studies with marker genes in more detail. Some of your species were already grouped in different subgenera.
4. *P. californicum*, with a differing alkaloid profile, is a New World species, whereas most others come from the Old World. The different origins and evolutionary scenarios need to be discussed.
5. Compare the timing of the split with known other splits in plants. Do we have similar timings in other groups with OW/NW disjunction?
6. Family names are not printed in italics

Reviewer #2:

Remarks to the Author:

In the present study by Catania et al under review in Nature communication, titled, "A functionally conserved STORR gene fusion in *Papaver* species that diverged 16.8 million years ago", authors described comparative genome analysis to understand origin of STORR gene fusion, a key event towards the evolution of BIAs, particularly towards promorphinans and morphinans biosynthesis.

Authors used a subset of 10 *Papaver* species, established transcriptome assemblies, genome assemblies, performed metabolome analysis, and functionally characterized STORR gene from selected *Papaver* species to reveal if it possesses the same function as that in Opium poppy.

I like the intension, design, and approach for this study, but the conclusion based on the data presented in this study is over statement for me, and I am not convinced. I have several comments on the kind of data acquired, and their interpretations, and I am listing them below-

1. To my surprised, authors decided not to include *P. somniferum* and close species, *P. setigerum* and *P. rhoeas* genomes reported by Yang et al in this study to draw a conclusion. They have probably two lines saying that Yang et al. did not consider the option of gene loss in the last paragraph of this study. But in fairness, the genome assemblies reported by Yang et al., including of Opium poppy is way way impressive than what authors have reported in this study. All 9 *Papaver* species genome that authors reported are fragmented, with very poor contiguity. In fact, Yang et al were able to assign unplaced scaffolds to the chromosomes of Opium poppy, and therefore, unless this study would have improved Poppy genome, I see absolutely no reason as to why not to use the better genome for comparative genome analysis. In my understanding, one possible reason could be the variant of species used in these studies, but for comparative genome analysis, authors need to include these three species together with what they have sequenced to draw any conclusion and any new insight into BIA evolution.
2. Authors need to include BUSCO score with Supplementary table S8. I understand that the genome size is close to what they achieved, but then there are so many other things that BUSCO offers insight into in terms of genome quality. This is a must for me.
3. The quality of the 9 genome assemblies reported in this study is poor. In terms of Contig N50, except *P. bracteatum* (Contig N50 is 1.4Mb), the rest has contig N50 from 2.8Kb (*P. nudicaule*) to 105Kb (*P. atlanticum*) for genomes sequenced using 10x. Authors have used synteny as one of the bases to look for collinearity, and that is one of the bases for their final conclusion. With this poor genome assembly, one could simply question the correctness of the genome and hence the synteny that they reported. While I am not questioning the data, the strong conclusion drawn through this study is still speculative to me and will need the better dataset to say things confidently.

4. Authors used transcriptome analysis to identify conserved orthogenes across 10 species and then used for drawing gene trees. This gene tree has served as the basis to explore the loss or presence of STORR or individual modules (p450 or oxidoreductase) across species. I wonder if authors should choose Iso-seq when they want to establish the transcriptome dataset for these 10 species. I understand the cost included, but then using Iso-seq will ensure that authors have full-length transcripts for all species, and in that case, the comparative analysis would make more sense with higher confidence. Illumina-based approach has a lot of limitations. Since the heterozygosity and complexity of genomes are different, the unigenes that were derived for these lines are going to have non-biological influence, and I wonder if that would affect the conclusion. I would have felt more comfortable if the gene tree analysis and ortho-analysis would have been done using annotated genome assemblies and not using transcriptome assemblies. Even if genomes are poor and fragmented, I feel that these would be better in terms of accurately representing gene sets with individual species reported here, and hence, will be more reliable to draw conclusion.

5. Among plant species that were sequenced in this study includes *P. bracteatum*, *P. armeniacum*, *P. nudicaule*, *P. californicum*, *P. atlanticum*, and *P. armeniacum*. On the other hand, for transcriptome assembly, species analyzed includes *P. pavoninum*, *P. nudicaule*, *P. californicum*, *P. atlanticum*, *P. orientale*, *P. bracteatum*, *P. triniifolium*, *P. armeniacum*, and *P. dubium*. Here the names in underlines are the species that were not used for genome sequencing. I am curious as to what was the basis for not sequencing these lines. In figure 1b, the authors reported no detection of promorphinans and morphinans biosynthesis for *Papaver dubium*, which is placed in the clade 2. Therefore, for me, genome sequencing for *P. dubium* would have been extremely important as then I could be able to look for specific genes that were lost, the regions where it got lost, and if homologs are present, any role of transposons that may result in no expression. Transcriptome-based assemblies just represent the expressed transcripts, and genes may not assemble even if present if expressed low at a specific time of tissue collection. That is another reason to use genome assemblies for gene tree construction in my opinion.

6. Another concern is about metabolite profiling. The authors seem to have reported the presence and absence of these metabolites (Fig1b). Did authors perform absolute quantification using approaches such as MRM since they have standards, and if not, why not? Is it not possible that the metabolites that seem not identified are actually in low abundance? I understand that some of these could not be dominant phytochemicals, but the presence of even a small amount would mean the genesets are present to carry on the biosynthesis. Therefore, quantification using a more sensitive high-resolution mass-spec is needed for metabolome analysis in this case. Here, relative quantification is not sufficient.

7. I am satisfied with Figure 2 where authors reported functional characterization of STORR homologs from four *Papaver* species. I am curious as to how to explain different levels of activity for STORR genes in these four species. I mean, why *Pca_STORR* have such a low activity for STORR compared to others? Why *Pso_STORR* have the highest activity? Something to do with the linker? It will benefit the readers if authors offer insights here.

To conclude, the evidence offered in this study is not strong enough to make such bold statements and interpretations. The datasets used keep the alternate hypothesis open. For example, authors say that analyzing *P. californicum* showed no promorphinans and genes associated with its biosynthesis, but the STORR gene is functional. If that's the case, why? I wonder if absolute quantification could detect a low accumulation of promorphinans in the *P. californicum*. And the genome assembly, probably the low-quality genome assembly and Illumina-seq based transcriptome assembly may be the reason for not detecting any genes. I feel that more strong datasets are needed to make such bold statements as done by the authors. Also, the quantity of genome assembly, which is super fragmented, force me to question synteny analysis and the basis on which conclusion was drawn. I am fascinated with the study but not convinced with the conclusion. Also, I recommend authors to include other published *Papaver* genome assemblies here (for example ones described by Yang et al), as these genome qualities are good and authors will be able to find strong evidence to support their hypothesis.

Reviewer #3:

Remarks to the Author:

The authors have presented a study of the evolution and function of the STORR gene in the papavers using de novo sequencing and various other omics approaches. The primary point of interest seems to be correcting a paper recently published in Nature Communications that showed evidence in support of STORR being relatively recently evolved. This paper instead presents evidence that STORR evolved much earlier, likely 16.8-25 MYA. Finding full-length copies of STORR in multiple species across the phylogeny provides pretty clear evidence for an earlier evolution, unless similar fusions evolved multiple times, which seems highly unlikely. While it is important to correct any inaccuracies presented by previous research, this work does not seem sufficiently interesting to merit publication in Nature Communications. It seems better suited to a more specialized journal or as a much shorter technical comment explicitly linked to the previous paper, as I think only people that work on papavers will care much about any of the results discussed here.

Reviewer #4:

Remarks to the Author:

This manuscript seeks to analyze the evolutionary origin of a gene fusion event to form STORR that led to the formation of (R)-reticuline from (S)-reticuline in the Papaveraceae. This is significant in the evolution of the morphine biosynthetic pathway, as the fusion protein provides an enzymatic entry point to the (R)-configured morphinan alkaloids. The authors use varied combinations of phylogenetic, transcriptomic, metabolomic, enzymatic and genomic analyses of thirteen members of the Papaveraceae (specifically *Papaver armeniacum*, *Papaver atlanticum*, *Papaver bracteatum*, *Papaver californicum*, *Papaver dubium*, *Papaver nudicaule*, *Papaver orientale*, *Papaver pavoninum*, *Papaver somniferum*, *Papaver triniifolium*, *Eschscholzia californica*, *Macleaya cordata* and *Aquilegia coerulea*). The study is logically designed and thoroughly executed. The authors conclude that the STORR gene fusion occurred once 16.8-25 mya, prior to the separation of *P. californicum* from other *Papaver* species. As *P. californicum* does not accumulate morphinan alkaloids, the STORR fusion does not exclusively lead to production of morphinan alkaloids in *Papaver*. The conclusions from Figure 4 in this study are also consistent with earlier targeted transcriptomic analyses that suggested that alkaloid profiles (in particular, the inability to produce morphinans) within *Papaver* spp. may have resulted from gene loss. The conclusions in this study are not consistent with a recently published study (Yang, X., Gao, S., Guo, L. et al. Three chromosome-scale *Papaver* genomes reveal punctuated patchwork evolution of the morphinan and noscapine biosynthesis pathway. *Nature Commun.* 12, 6030 (2021). <https://doi.org/10.1038/s41467-021-26330-8>) that does not consider gene loss in interpretation of the data. As more and more alkaloid pathways are elucidated at the genome level, studies of this type will provide important insight into the evolutionary origins, and possible functions, of these metabolites.

Suggestion: The major metabolite analysis data presented in Table S1 is understandably incomplete, but it would be more useful if high resolution mass-to-charge ratios can be provided for each unknown major metabolite in latex/capsule of *P. dubium*, *P. nudicaule*, *P. pavoninum* and *P. triniifolium*.

Invited Re-Submission of Nature Communications manuscript NCOMMS-21-42033-T

Catania *et al.*, "A functionally conserved *STORR* gene fusion in *Papaver* species that diverged 16.8 million years ago"

Response to Reviewers

In the following text all reviewer comments are italicized, author responses are non-italicized, figures/tables citations are in bold and revisions in excerpts from main text are highlighted in red.

Finally, we would like to thank the reviewers and yourself for all the constructive and positive feedback on our manuscript.

Response to reviewer's comments:

REVIEWER COMMENTS

Reviewer #1 (Remarks to the Author):

*This is an interesting study on the evolution of the *STORR* gene in the genus *Papaver*. In general, the ms is well written and the methods used for analysis are adequate. The following minor issues need attention.*

1. make clear what you consider a morphinan alkaloid

We thank reviewer 1 for pointing out the confusion regarding what we consider a morphinan alkaloid and we hope that the alteration to the text on page 4 will provide clarification on what we are considering a morphinan.

On page 4: "The benzylisoquinoline alkaloids or BIA's represent a structurally diverse group predominantly identified in the order Ranunculales^{1,2}. **The naturally synthesised morphinans thebaine, oripavine, codeine and morphine are part of the BIA class of alkaloids, with morphine renowned for its powerful analgesic properties.** They are naturally synthesised in the genus *Papaver* and are currently commercially produced in opium poppy, *Papaver somniferum*, from the Papaveraceae family. ..."

*2. Only morphine is a strong analgesic within morphinans. It is only produced by *P. somniferum*. The sentence in the ms is misleading.*

We thank reviewer 1 for drawing our attention to the sentence which is misleading. We hope that the clarification to the text on page 4 has resolved the problem.

On page 4: " The benzylisoquinoline alkaloids or BIA's represent a structurally diverse group predominantly identified in the order Ranunculales^{1,2}. **The naturally synthesised morphinans thebaine, oripavine, codeine and morphine are part of the BIA class of alkaloids, with morphine renowned for its powerful analgesic properties.** They are naturally synthesised in the genus *Papaver* and are currently commercially produced in opium poppy, *Papaver somniferum*, from the Papaveraceae family. ..."

3. *if you discuss the phylogeny of Papaver, you might discuss previous studies with marker genes in more detail. Some of your species were already grouped in different subgenera.*

We have made the following changes in the main text based on the reviewer's comment:

On pages 6-7: "... The topology of the tree generated based on the species included is largely congruent with other phylogenetic trees constructed based on taxonomic sequence datasets of chloroplast, ribosomal and plastid markers from the genus *Papaver* and the wider family and order^{24,25,26,27,28}. The differences observed in the ordering of the *Papaver* species can be attributed to the marker sets and methods used for the assembly of the species tree and range of plant samples sequenced. The divergence times estimated between the species of the tree are in the regions of those previously estimated for example the divergence of *P. californicum* at around 16.8 million years ago (MYA) compares to previously estimated timings.^{27,28,29,30} ..."

4. *P. californicum, with a differing alkaloid profile, is a New World species, whereas most others come from the Old World. The different origins and evolutionary scenarios need to be discussed.*

We thank reviewer 1 for drawing our attention to the different origins and evolutionary scenarios relating to *P. californicum*, disjunction of *P. californicum* from the other *Papaver* species and its evolutionary origins. We have made the following additions to the text on page 12, to incorporate information on the origins and evolutionary scenarios regarding *P. californicum*.

On pages 11-12: "... *P. californicum* is an 'Old World' *Papaver* species indigenous to California thought to have evolved in parallel to Eurasian members of the family^{24,27,29,30}. The specific distribution of *P. californicum* to North Western America is an example of an 'Old World/New World' disjunction at 28 - 10 MYA in the Papaveraceae distribution³⁰. The occurrence of such North American/Eurasian disjunctions is recognised in a number of species, and attributed to historic changes in climate and existence of previous land-bridge connections^{30,31,32,33}. The differences observed in the metabolite profile of *P. californicum* compared to the morphinan producing species could be attributed to its parallel evolution in a different environment with different selective pressures...."

5. *Compare the timing of the split with known other splits in plants. Do we have similar timings in other groups with OW/NW disjunction?*

It is an interesting point to look at all the species that are affected by disjunctions and the timings that occurred, however we feel that to include further full discussions on this point would be confusing based on our focus on the evolution of the STORR gene,

In response to point 4 of reviewer 1 above we have attempted to address this point appropriately with the adjusted text on page 12 with several relevant publications being cited.

6. *Family names are not printed in italics*

We thank the reviewer for pointing this out, it is now corrected throughout the manuscript.

Reviewer #2 (Remarks to the Author):

In the present study by Catania et al under review in Nature communication, titled, "A

functionally conserved STORR gene fusion in Papaver species that diverged 16.8 million years ago”, authors described comparative genome analysis to understand origin of STORR gene fusion, a key event towards the evolution of BIAs, particularly towards promorphinans and morphinans biosynthesis. Authors used a subset of 10 Papaver species, established transcriptome assemblies, genome assemblies, performed metabolome analysis, and functionally characterized STORR gene from selected Papaver species to reveal if it possesses the same function as that in Opium poppy.

I like the intension, design, and approach for this study, but the conclusion based on the data presented in this study is over statement for me, and I am not convinced.

We note reviewer 2 likes the intention, design and approach of our study. We have taken on board the recommendation regarding genome annotation of the five new genome assemblies and have collaborated with Professor Yves Van de Peer, University of Gent, an expert in comparative genome evolution and genome annotation to produce a comprehensive annotation of all five genomes - see detailed response below.

We wish to challenge the conclusion of reviewer 2 that the evidence we present is not strong enough to support the conclusions of our study and we will do so in our response to the reviewer's comments below.

I have several comments on the kind of data acquired, and their interpretations, and I am listing them below-

1. To my surprised, authors decided not to include P. somniferum and close species, P. setigerum and P. rhoeas genomes reported by Yang et al in this study to draw a conclusion. They have probably two lines saying that Yang et al. did not consider the option of gene loss in the last paragraph of this study. But in fairness, the genome assemblies reported by Yang et al., including of Opium poppy is way way impressive than what authors have reported in this study. All 9 Papaver species genome that authors reported are fragmented, with very poor contiguity. In fact, Yang et al were able to assign unplaced scaffolds to the chromosomes of Opium poppy, and therefore, unless this study would have improved Poppy genome, I see absolutely no reason as to why not to use the better genome for comparative genome analysis. In my understanding, one possible reason could be the variant of species used in these studies, but for comparative genome analysis, authors need to include these three species together with what they have sequenced to draw any conclusion and any new insight into BIA evolution.

All of our initial data analyses and manuscript preparation were completed for submission before the work of Yang *et al* had been published and a pre-print of our manuscript was published in BioRXIV prior to the Yang *et al* publication.

Having carefully considered the content of Yang *et al* we made the decision that inclusion of the updated *P. somniferum* genome assembly and the additional two *Papaver* genome assemblies in our manuscript was not necessary as they would not change the main conclusions therein. That said, we have now taken the opportunity of this revision to include the additional *P. setigerum* and *P. rhoeas* genomes in all our analyses. The inclusion and consideration of these additional data has not altered any of the conclusions reached in the first version of our manuscript. The inclusion of the additional *Papaver* genome data has in fact strengthened our conclusions regarding the evolution of STORR as they clearly demonstrate the loss of STORR in the *P. rhoeas* genome. Furthermore the inclusion of the *P. setigerum* and *P. rhoeas* allows us to further substantiate the earlier evolution of the STORR gene than is being proposed in Yang *et al*.

The following figures, supplemental tables and supplemental datasets have been modified to include the extra information from the two additional genomes:

- **Figure 1c** (species phylogeny)

- **Figure 2a** (alignment of the linker region of STORR proteins)
- **Figure 3a & 3b** (gene trees of CYP82Y2-L and COR-L subfamilies) and **Figure 3c** (Schematic representation of the evolutionary history of *STORR*)
- **Figure 4a** (synteny between contigs that contain *STORR* and pro-morphinan genes in *Papaver* genomes) and **Figure 4b** (evolutionary history of clustering of the *STORR* and pro-morphinan genes)
- **supplemental Figure 1** (alignment of *STORR*)
- **supplemental table S5** (BIA genes in *Papaver* species)
- **supplemental table S6** (list of COS genes)
- **supplemental table S7** (% ID matrix between *STORR* homologs)
- **supplemental table S9** (Synteny analysis)
- **supplemental dataset 2** (COS gene sequences)
- **supplemental dataset 3 and 4** (nucleotide and protein sequences of *STORR*)
- **supplemental datasets 6, 7 and 8** (coding sequences of *STORR* and all CYP82Y2-L, COR-L members and genomic sequences of *STORR* and CYP82Y2-L/COR-L gene pairs in **Figure3**)
- **supplemental datasets 9** (syntenic regions corresponding to the BIA gene cluster in opium poppy)

Relevant changes have been made throughout the manuscript to highlight the inclusion of the additional *P. setigerum* and *P. rhoeas* genomes; including:

On page 5: "... .To investigate the presence of the *STORR* gene fusion across the *Papaver* genus we selected nine other *Papaver* species that provided good taxonomic coverage (**Fig.1b,1c; supplemental table S1**), and analysed them alongside the reported metabolite and whole genome assembly data of opium poppy^{5,14}, *P. rhoeas* and *P. setigerum*¹⁴. ..."

On page 9: "... Our analysis of *STORR* gene evolution also reveals the importance of lineage specific deletion (*P. atlanticum* and *P. rhoeas*), duplication (*P. californicum*, *P. armeniacum*, *P. somniferum* and *P. setigerum*) and rearrangement after duplication (*P. californicum*) (**Fig.3c**). In addition, we note that in *P. somniferum* one copy of *STORR* has been lost after a whole genome duplication event⁵. ... "

On page 10: "... The order of the P450 and oxidoreductase genes for both paralogous pairs in *P. armeniacum* and *P. rhoeas*, as well as two of the three gene pairs in *P. setigerum*, have switched compared to the other species (**Fig.3c**). Taken into consideration gene deletion, duplication, rearrangement and erosion it is still apparent from these findings that paralogous pairing of a CYP82Y2_La and COR_La existed in the clade 2 common ancestor (**Fig.3c**), supporting the hypothesis that a segmental duplication would have occurred prior to *STORR* fusion/neofunctionalization. ..."

On page 11: "... As part of the 800kb BIA cluster, *STORR* is clustered with four promorphinan genes in the opium poppy genome, which also contains a **second** syntenic region containing paralogs of the promorphinan genes but not *STORR*⁵ (**Fig.4a**). **Both regions are well conserved in *P. setigerum*, the closest related sister species of opium poppy, with one on chromosome 15 and the other on chromosome 8 (Fig.4a; supplemental table S9)**. Both regions contain *STORR* and all four pro-morphinan genes, with an extra tandem duplicated copy of *SALAT* present in the region on Chromosome 8. We found that *P. bracteatum* shows very good synteny with the opium poppy regions containing the pro-morphinan genes with a notable difference being an extra copy of *SALR* in the former (**Fig.4a; supplemental table S9**)...."

2. Authors need to include BUSCO score with Supplementary table S8. I understand that the genome size is close to what they achieved, but then there are so many other things that BUSCO offers insight into in terms of genome quality. This is a must for me.

We thank the reviewer for this comment.

In collaboration with the group of Professor Yves Van de Peer, University of Gent all five genomes have now been annotated and submitted to NCBI. Prof Van de Peer and Jiyang Chang are included as co-authors on our revised manuscript. A summary of the genome annotations including the BUSCO score is included in the updated **supplemental table S8** with details of the annotation process in the updated methods section.

The BUSCO scores of the genomes *P. californicum*, *P. atlanticum*, *P. bracteatum*, and *P. armeniacum* are all over 94% indicating excellent genome coverage. *P. nudicaule* has a lower (71%) BUSCO score as had been expected on the basis of the high level of heterozygosity in this species. To overcome this issue in *P. nudicaule* we used the initial gene hits from the genome searches for the CYP82Y2-like and COR-like sequences to guide 5' and 3' extension and PCR amplification to produce full length genomic sequences directly from the same plant material as used for genomic sequencing as described in the methods section.

3. The quality of the 9 genome assemblies reported in this study is poor. In terms of Contig N50, except P. bracteatum (Contig N50 is 1.4Mb), the rest has contig N50 from 2.8Kb (P. nudicaule) to 105Kb (P. atlanticum) for genomes sequenced using 10x. Authors have used synteny as one of the bases to look for collinearity, and that is one of the bases for their final conclusion. With this poor genome assembly, one could simply question the correctness of the genome and hence the synteny that they reported. While I am not questioning the data, the strong conclusion drawn through this study is still speculative to me and will need the better dataset to say things confidently.

We agree with the reviewer that high contiguity in genome assemblies is essential for global comparison of genomes. However, our synteny analysis in **Figure 4** focusses on the comparison of the region surrounding the *STORR* gene in different *Papaver* species and its relation to orthologs of other promorphinan genes of the opium poppy BIA gene cluster. We do not make any claims based on whole genome assembly comparisons in this manuscript. The synteny that we do report in **Figure 4** is based on single contigs and does not depend on the overall contiguity of the respective genome assemblies. For example, the *P. bracteatum* and *P. armeniacum* BIA cluster regions shown in **Figure 4a** are based on evidence from single contigs. Improving the genome assembly would have little or no effect on the quality of the evidence presented in **Figure 4**.

In addition, it is most important to note that the BIA genes associated with the *STORR* syntenic region for each species presented in **Figure 4a** are consistent with BIA gene presence in the transcriptomic data for each of these species.

4. Authors used transcriptome analysis to identify conserved orthogenes across 10 species and then used for drawing gene trees. This gene tree has served as the basis to explore the loss or presence of STORR or individual modules (p450 or oxidoreductase) across species. I wonder if authors should choose Iso-seq when they want to establish the transcriptome dataset for these 10 species. I understand the cost included, but then using Iso-seq will ensure that authors have full-length transcripts for all species, and in that case, the comparative analysis would make more sense with higher confidence. Illumina-based approach has a lot of limitations. Since the heterozygosity and complexity of genomes are different, the unigenes that were derived for these lines are going to have non-biological influence, and I wonder if that would affect the conclusion. I would have felt more comfortable if the gene tree analysis and ortho-analysis would have been done using annotated genome assemblies and not using transcriptome assemblies. Even if genomes are poor and fragmented, I feel that these would be better in terms of accurately

representing gene sets with individual species reported here, and hence, will be more reliable to draw conclusions.

We agree completely with the reviewer that gene tree analysis should be carried out using genome assemblies and full length coding sequences should be used where possible to improve confidence. This is in fact the approach that we have taken in our gene tree analyses of the STORR modules. In response to the suggestion from reviewer 2 above we have also now added the *P. setigerum* and *P. rhoeas* genome assemblies in addition to the *P. somniferum* and five new genome assemblies reported in the current study.

It is important to note that we did not rely on transcriptome assemblies for the gene tree analysis which seems to have been suggested by reviewer 2. We started with blast searches using genome assembly sequences and all potential sequence regions were followed by close examination of gene structures based on homology. Therefore, great care had been taken to validate each sequence and to ensure full coding sequence representation for all sequences included in **Figure 3** with the exception of one single CYP82Y2-L sequence in the outgroup. We also have now used the fully annotated gene datasets of the five new genome assemblies to confirm the gene sequences identified following the above approach. All sequences described are represented in **supplemental datasets 6-8**.

We consider that our approach outlined above is actually more sensitive, comprehensive and inclusive than just relying on a whole genome annotated gene set. The approach used has produced full length gene sequences from genome assemblies with a high degree of coverage as evidenced by the BUSCO analysis.

We therefore think that there would be little or no benefit in adding an Iso-seq dataset to this study.

We have modified the following relevant text to make it clear that we have used gene sequence data from whole genome assemblies for the gene tree analysis. In addition, as noted above, we have also incorporated the data from the two additional genomes *P. rhoeas* and *P. setigerum* from Yang *et al* (2021) into the current study.

Figures 3a, 3b, and 3c, supplemental tables S5, S7 and supplemental datasets 6-8 have been modified accordingly.

On page 9: "...In order to establish if equivalent paralogous pairs are present in related *Papaver* species we used whole genome sequencing approaches to assemble draft genomes for *P. nudicaule* from clade 1 and *P. californicum*, *P. bracteatum*, *P. atlanticum*, and *P. armeniacum* from clade 2 (**supplemental table S8**). We compiled and annotated all homologous sequences that contain full length genes corresponding to either of the STORR modules in these draft assemblies (**supplemental datasets 6&7**). **We combined all CYP82Y2 and oxidoreductase sequences from the five new genomes with those** retrieved from searches of the annotated opium poppy^{5,14}, *P. rhoeas* and *P. setigerum*¹⁴ genomes, as well as the transcriptomic data **of other species in** the present study. **We then** constructed gene trees for the two gene subfamilies containing the coding sequences closely related to the STORR modules CYP82Y2 (**Fig.3a**) and oxidoreductase (**Fig.3b**) respectively. ..."

5. Among plant species that were sequenced in this study includes *P. bracteatum*, *P. armeniacum*, *P. nudicaule*, *P. californicum*, *P. atlanticum*, and *P. armeniacum*. On the other hand, for transcriptome assembly, species analyzed includes *P. pavoninum*, *P. nudicaule*, *P. californicum*, *P. atlanticum*, *P. orientale*, *P. bracteatum*, *P. triniifolium*, *P. armeniacum*, and *P. dubium*. Here the names in underlines are the species that were not used for genome sequencing. I am curious as to what was the basis for not sequencing these lines. In figure 1b, the authors reported no detection of promorphinans and morphinans biosynthesis for

Papaver dubium, which is placed in the clade 2. Therefore, for me, genome sequencing for *P. dubium* would have been extremely important as then I could be able to look for specific genes that were lost, the regions where it got lost, and if homologs are present, any role of transposons that may result in no expression. Transcriptome-based assemblies just represent the expressed transcripts, and genes may not assemble even if present if expressed low at a specific time of tissue collection. That is another reason to use genome assemblies for gene tree construction in my opinion.

Reviewer 2 asks on what basis species were selected for genome sequencing and why *P. dubium* was not included. We took metabolite data as well as taxonomic coverage into account when selecting a subset of species for genome sequencing. We have now included *P. rhoeas* and *P. setigerum* from Yang *et al.* (2021) in our revised study. Analysis including the Yang *et al* data shows that *P. rhoeas* (which also lacks the STORR gene) is a sister of *P. dubium* (**Figure 1C**). Thus, the inclusion of *P. rhoeas* in our analysis can help address the point raised by reviewer 2 regarding *P. dubium*.

6. Another concern is about metabolite profiling. The authors seem to have reported the presence and absence of these metabolites (Fig1b). Did authors perform absolute quantification using approaches such as MRM since they have standards, and if not, why not? Is it not possible that the metabolites that seem not identified are actually in low abundance? I understand that some of these could not be dominant phytochemicals, but the presence of even a small amount would mean the genesets are present to carry on the biosynthesis. Therefore, quantification using a more sensitive high-resolution mass-spec is needed for metabolome analysis in this case. Here, relative quantification is not sufficient.

We thank the reviewer for raising this important point. **Figure 1b** is a representation of the quantitative data of latex and capsule extracts fully presented in **supplemental table 2**. The figure has been amended so as not to describe presence and absence but rather presence and not detected (ND). Confirming a true negative by absence of detection is always challenging, for any analytical technique. We deployed a commonly used technique for untargeted analysis, namely High Resolution Accurate Mass (HRAM) LC-MS, using an Orbitrap instrument in full-scan mode.

We do have access to appropriate morphinan reference standards and have incorporated these into our HRAM workflows. This hybrid technique enables us to use full-scan MS data to discover unknowns while still quantifying known compounds based on three criteria: accurate mass (< 5ppm), retention time matching, and from the calibration curves of authentic standards. Crucially, we have determined limits of detection (LOD) from linear standard curves on our Orbitrap using the $3\sigma/S$ method (where σ is the standard deviation of the response, and S the slope).

We assume that these comments regarding sensitivity of metabolite analysis may be based on the report in the Yang *et al* 2021 publication that trace amounts of morphinans were detected in *P. rhoeas* but no morphinan related genes including *STORR* were present in the assembled genome of this species. We note that the detection method used by Yang *et al* did not include any morphinan standards which calls into question the identify of the assigned morphinan peaks. We briefly address this matter in our revised manuscript as detailed below.

On Pages 5-6 "... **These two species along with *P. somniferum* and *P. setigerum*¹⁴ were the only species identified as producing morphinans. A recent report of trace amounts of morphinans in *P. rhoeas* appears to have been conducted without the use of known morphinan standards¹⁴. Our analysis of *P. rhoeas* capsule material conducted using High Resolution Accurate Mass LC-MS does not detect morphinans or promorphinans above defined limits of detection (**Supplemental table S2**), which for morphine is 226-fold lower than morphine levels in *P. setigerum*. Our results of zero peak area for**

morphinans in *P. rhoeas* are in agreement with other published results^{17,22,23} and consistent with the absence of morphinan related genes in this species. ...”

7. I am satisfied with Figure 2 where authors reported functional characterization of STORR homologs from four *Papaver* species. I am curious as to how to explain different levels of activity for STORR genes in these four species. I mean, why *Pca*_STORR have such a low activity for STORR compared to others? Why *Pso*_STORR have the highest activity? Something to do with the linker? It will benefit the readers if authors offer insights here.

The aim of our analysis of STORR protein activity in microsomal preparations of *S. cerevisiae* extracts was to establish substrate specificity which we have done. We did use the same amount of protein in the assays but we would prefer to be cautious about directly comparing the relative abundance of (*R*)-reticuline produced in the different assays and extrapolating this to specific activity *in-planta* as the different *S. cerevisiae* cultures could be expressing different amounts of STORR protein. There is of course the possibility that STORR has evolved to exhibit different specific activities in the four plant species we have shown to express a functional STORR but we would prefer not to speculate about this in the manuscript.

To conclude, the evidence offered in this study is not strong enough to make such bold statements and interpretations. The datasets used keep the alternate hypothesis open. For example, authors say that analyzing P. californicum showed no promorphinans and genes associated with its biosynthesis, but the STORR gene is functional. If that's the case, why? I wonder if absolute quantification could detect a low accumulation of promorphinans in the P. californicum. And the genome assembly, probably the low-quality genome assembly and Illumina-seq based transcriptome assembly may be the reason for not detecting any genes. I feel that more strong datasets are needed to make such bold statements as done by the authors. Also, the quantity of genome assembly, which is super fragmented, force me to question synteny analysis and the basis on which conclusion was drawn. I am fascinated with the study but not convinced with the conclusion. Also, I recommend authors to include other published Papaver genome assemblies here (for example ones described by Yang et al), as these genome qualities are good and authors will be able to find strong evidence to support their hypothesis.

We have addressed these views of reviewer 2 in our detailed responses above apart from the point regarding STORR function in *P. californicum*. We do address this point on page 13 of the main text when we discuss the possible route to biosynthesis of the most abundant BIA in *P. californicum*, glaucine. We had included the sentence: ‘Evidence for (*S*)-glaucine biosynthesis via (*S*)-reticuline has previously been proposed²⁷. It is therefore interesting to speculate, based on the emergence of STORR at 16.8 MYA in the *Papaver* lineage, that the *P. californicum* STORR could be involved in (*R*)-glaucine formation.’

In summary: all of our conclusions are supported by robust evidence:

- The STORR gene fusion is present in six *Papaver* species that diverged from a common ancestor 16.8 MYA.
- The evolutionary history of STORR is backed up by gene tree analysis of the full-length members of *CYP82Y2-L* and *COR-L* subfamilies that include the STORR modules. These sequences were derived from assembled genomes with high degrees of genome coverage as demonstrated by BUSCO scores. The addition of data from the *P. setigerum* and *P. rhoeas* genomes, as suggested by reviewer 2, did not affect our conclusion. In fact, there is lack of evidence for the claim made by Yang *et al* that the STORR fusion event occurred after the WGD event at 7.2 MYA in opium poppy. This is because the hypothesis of Yang *et al* would associate the segmental gene duplication giving rise to the *CYP82Y2-La/COR-La* gene pair and STORR with the opium poppy WGD event, which is problematic given the data

presented in the current manuscript as well as in earlier report of a functional *STORR* gene being present in *P. bracteatum* with that work cited in our manuscript.

We have modified the main text to further clarify this point.

- Our conclusion of the emergence of gene clustering of *STORR* and pro-morphinan genes is supported by the presence of the gene cluster in a contiguous single sequence in the *P. bracteatum* genome in **Figure 4a**.

Relevant text changes:

On page 14: "... A recent report based on genome comparison of opium poppy with two very closely related *Papaver* species, *P. setigerum* and *P. rhoeas* has proposed the fusion event that resulted in *STORR* occurred following the whole genome duplication event in opium poppy after its divergence from *P. rhoeas*¹⁴. However, that proposal does not consider the possibility of *STORR* gene loss within *P. rhoeas* nor does it take into account the previously reported presence of *STORR* in *P. bracteatum*⁷ which diverged earlier from all three species (**Fig.1c**)²⁴. Furthermore, that proposal associates the segmental duplication giving rise to the CYP82Y2-La/COR-La gene pair and *STORR* with the opium poppy WGD event but no supporting evidence was presented¹⁴. Figure 3 demonstrates that the required gene duplication event must have happened before this WGD event. ..."

Reviewer #3 (Remarks to the Author):

The authors have presented a study of the evolution and function of the STORR gene in the papavers using de novo sequencing and various other omics approaches. The primary point of interest seems to be correcting a paper recently published in Nature Communications that showed evidence in support of STORR being relatively recently evolved. This paper instead presents evidence that STORR evolved much earlier, likely 16.8-25 MYA. Finding full-length copies of STORR in multiple species across the phylogeny provides pretty clear evidence for an earlier evolution, unless similar fusions evolved multiple times, which seems highly unlikely. While it is important to correct any inaccuracies presented by previous research, this work does not seem sufficiently interesting to merit publication in Nature Communications. It seems better suited to a more specialized journal or as a much shorter technical comment explicitly linked to the previous paper, as I think only people that work on papavers will care much about any of the results discussed here.

Reviewer 3 agrees with our findings and states that it is important to 'correct any inaccuracies presented by previous research' but then questions whether it should be published in Nature Comms.

We think it most definitely should for a number of reasons.

The *STORR* gene fusion event is considered a key step in the evolution of benzyloisoquinoline alkaloid (BIA) metabolism in opium poppy as the resulting bi-modular protein performs the isomerization of (S)- to (R)-reticuline which is required for morphinan biosynthesis. Our previous analysis of the opium poppy genome (Guo *et al.*, 2018) suggested the fusion event occurred before a whole genome duplication event between 7 and 8 million years ago. From the literature we knew that at least one other *Papaver* species also contained a *STORR* gene fusion which raised questions about how widespread the gene fusion is across *Papaver* species, is it exclusively associated with morphinan biosynthesis, when and how often did the fusion event occur?

In the current manuscript we use a combination of phylogenetic, transcriptomic, metabolomic, biochemical and genomic analysis to investigate the origin of the *STORR* gene fusion and *STORR* function across the Papaveraceae family. We now show that the pro-

morphinan/morphinan subclass of BIAs is present in a subset of 12 *Papaver* species including *P. somniferum* (opium poppy) and this correlates with the presence of the *STORR* gene fusion with *one important exception*. *P. californicum* does not produce morphinans but it does contain a *STORR* gene fusion that epimerizes (S)- to (R)-reticuline when heterologously expressed in yeast. We discovered that the most abundant BIA in *P. californicum* is (R)-glaucine, a member of the aporphine subclass of BIAs. Only the (S) isomer of this compound has previously been reported from nature and is commercially isolated from *Glaucium* species in the Papaveraceae and used medically in some countries as an antitussive agent for cough treatment. (R)-glaucine has been chemically synthesised by others and also has interesting bioactivity. Our discovery of its presence in *P. californicum* and association with *STORR* lays the foundation to elucidate the biosynthetic pathway of these important compounds.

The high similarity of the amino acid sequence linking the two modules of *STORR* from all four species along with phylogenetic gene tree analysis demonstrate that the gene fusion occurred only once and between 16.8-24.1 million years ago before the separation of *P. californicum* from the other *Papaver* species. These results lead us to conclude that the function of the *STORR* gene fusion is not exclusive to morphinan production in the Papaveraceae.

Our work overturns claims made in the Yang *et al.*, 2021 Nature Communications publication that the *STORR* gene fusion occurred after the whole genome duplication event in opium poppy. That work was based on genome comparison of opium poppy with two very closely related *Papaver* species, *P. setigerum* and *P. rhoeas*. However, that proposal does not consider the possibility of *STORR* gene loss within *P. rhoeas* nor does it take into account the previously reported presence of *STORR* in *P. bracteatum* which diverged earlier from all three species used in that study. We constructively address this matter in the current revised manuscript. The revision includes incorporation of the *P. rhoeas* and *P. setigerum* genome assemblies from Yang *et al.* as detailed above. Inclusion of this genomic data in fact supports the conclusions of our original submission and provides good evidence that the *STORR* gene has been lost in the *P. rhoeas* lineage.

The role of gene clustering, genome duplication, deletion and rearrangement in the evolution of plant specialized metabolism is currently a very active area of research that is benefitting significantly from the availability of whole genome assemblies. Our manuscript shows the value of adopting a multidisciplinary approach to clearly define an evolutionary series of events that can lead to new insights into BIA metabolism. This work from our laboratory builds on a series of high-profile publications describing gene clustering, gene fusion and genome rearrangement in opium poppy. With this publication we are establishing BIA metabolism as a model system for studies into the evolution of specialised metabolism in higher plants and as such should lead to increased interest and research in the area.

For all these reasons we think it merits publication in Nature Communications.

Reviewer #4 (Remarks to the Author):

This manuscript seeks to analyze the evolutionary origin of a gene fusion event to form STORR that led to the formation of (R)-reticuline from (S)-reticuline in the Papaveraceae. This is significant in the evolution of the morphine biosynthetic pathway, as the fusion protein provides an enzymatic entry point to the (R)-configured morphinan alkaloids. The authors use varied combinations of phylogenetic, transcriptomic, metabolomic, enzymatic and genomic analyses of thirteen members of the Papaveraceae (specifically Papaver armeniacum, Papaver atlanticum, Papaver bracteatum, Papaver californicum, Papaver dubium, Papaver nudicaule, Papaver orientale, Papaver pavoninum, Papaver somniferum,

Papaver triniifolium, *Eschscholzia californica*, *Macleaya cordata* and *Aquilegia coerulea*). The study is logically designed and thoroughly executed. The authors conclude that the STORR gene fusion occurred once 16.8-25 mya, prior to the separation of *P. californicum* from other *Papaver* species. As *P. californicum* does not accumulate morphinan alkaloids, the STORR fusion does not exclusively lead to production of morphinan alkaloids in *Papaver*. The conclusions from Figure 4 in this study are also consistent with earlier targeted transcriptomic analyses that suggested that alkaloid profiles (in particular, the inability to produce morphinans) within *Papaver* spp. may have resulted from gene loss. The conclusions in this study are not consistent with a recently published study (Yang, X., Gao, S., Guo, L. et al. Three chromosome-scale *Papaver* genomes reveal punctuated patchwork evolution of the morphinan and noscapine biosynthesis pathway. *Nature Commun.* 12, 6030 (2021). <https://doi.org/10.1038/s41467-021-26330-8>) that does not consider gene loss in interpretation of the data. As more and more alkaloid pathways are elucidated at the genome level, studies of this type will provide important insight into the evolutionary origins, and possible functions, of these metabolites.

Suggestion: The major metabolite analysis data presented in Table S1 is understandably incomplete, but it would be more useful if high resolution mass-to-charge ratios can be provided for each unknown major metabolite in latex/capsule of P. dubium, P. nudicaule, P. pavoninum and P. triniifolium.

Reviewer 4 provides a very positive and constructive review of our manuscript. Thanks for the useful suggestion to include more high resolution mass-to-charge ratio data in **Table S1**.

Table S1 has been amended to include the mass-to charge ratios for the unknown major metabolites of *P. dubium*, *P. nudicaule*, *P. pavoninum* and *P. triniifolium*.

Reviewers' Comments:

Reviewer #1:

Remarks to the Author:

Thank you for following my suggestions.

In Line 220 you call *P. californicum* and "Old World" species. As it only occurs in Americas, this should be a "New World species"

Reviewer #2:

Remarks to the Author:

In the present revision for the manuscript, titled, "A functionally conserved STORR gene fusion in *Papaver* species that diverged 16.8 million years ago", authors have provided new analysis by including two of the previously published genomes and rebuttal for several of my comments. Authors in the current version of the manuscript reanalyzed the entire dataset by including previously published genomes of *P. rhoeas* and *P. setigerum*. In the rebuttal, the authors mentioned that using the previously published Poppy genome (which is higher quality in terms of assembly contiguity) does not change the main conclusion of this manuscript. I accept the argument. I also note the fact this study was published as BioRxiv prior to the Yang et al publication.

After going through all the responses from the authors and the changes that were made while addressing all reviewers, I now find that this study in its present form missed to address many interesting analyses that would offer deep insights on the evolution of specialized metabolites and the mechanism post the emergence of STORR fusion event. The STORR fusion event and its importance has been previously described in detail. The only core message from this study emerges as that the origin of the STORR gene fusion across the *Papaveraceae* family occurred between 16.8-24.1 million years ago before the separation of *P. californicum* from the other *Papaver* species. In that sense, I agree with reviewer #4 comments in the first round of review, and I am not satisfied with the explanation that authors have provided. With the kind of data authors have generated, I strongly feel that authors could find footprints of how STORR fusion events shaped the present-day existing biosynthesis pathway. Without this, I find the correction of previously published Nature Communication article by incorporating more species is not enough.

One of my major criticisms of this manuscript as mentioned in several of my comments were the poor contiguity of the genome, for which, the author's rebuttal is as follows-

"We agree with the reviewer that high contiguity in genome assemblies is essential for global comparison of genomes. However, our synteny analysis in Figure 4 focusses on the comparison of the region surrounding the STORR gene in different *Papaver* species and its relation to orthologs of other promorphinan genes of the opium poppy BIA gene cluster. We do not make any claims based on whole-genome assembly comparisons in this manuscript. The synteny that we do report in Figure 4 is based on single contigs and does not depend on the overall contiguity of the respective genome assemblies. For example, the *P. bracteatum* and *P. armeniacum* BIA cluster regions shown in Figure 4a are based on evidence from single contigs. Improving the genome assembly would have little or no effect on the quality of the evidence presented in Figure 4."

For me, this then becomes a very specific question and I then find this article more suitable to a plant specific journal and not within the broad readership of Nature communication. This study will then be of interest to only those researchers who are interested in *papaver* species, especially using the generated data resource for further functional analysis. Other than the fact that we improved the estimated time of the evolution of STORR, I fail to see how the results are applied to explore the overall evolution of the biosynthesis pathways based on this knowledge. My reason for advocating on improving the genome assemblies was with the hope that a comprehensive comparative genome analysis could reveal several questions such as-

1. Does the evolution of STORR result in the loss or gain of a specific set of enzymes that are involved

in the biosynthesis pathway of core metabolites

2. Clearly *P. californicum* has functional STORR, but it does not produce several of the core metabolites that Poppy does. Why? Does it have to do with the gain or loss of some specific genes? A comprehensive genome-wide synteny analysis could provide such answers.
3. Does enzymes known to be involved in the pro-morphinan/morphinan subclass of BIAs biosynthesis followed the event of STORR fusion. Authors reported the rate of substitution in their previously published article on Poppy genome, and they could estimate an approximate time of emergence of key genes and judge their emergence with STORR.
4. Knowing the approximate timing of STORR, what enzymes were evolved or retained or gained around that time period within producing and non-producing species, does that has any influence on the evolution of the core metabolites?
5. Does the copy number of genes involved in the pro-morphinan/morphinan subclass of BIAs biosynthesis increase within producing plants compared to the ones that does not produces?
6. Does the event of STORR fusion had any role in shaping or reorganizing genomes of the producing plants?
7. Does plants producing pro-morphinan/morphinan show high degree of collinearity overall? How about the gene clusters?
8. As authors mentioned the fusion event of STORR does not mean that the plant will produce pro-morphinan/morphinan, then what is the determining factor? Is there an essential set of genes that one could propose? I am not sure if such a method exists, the fact remains that the STORR genes require key precursors for subsequent reactions downstream. Therefore, by comparing *P. californicum* with rest of the producing plants, and by including the plants which observed no fusion of STORR, one could find genes that evolved under influence of STORR fusion and a set of genes that further gained to offer present-day chemotypes.

These and several additional analyses and answers could be derived using the dataset that authors have generated in this study. In my opinion, the goal of this study needs to be broadened. Authors did report (R)-glauanine in *P. californicum*, and therefore, its genome becomes valuable. However, that's not the main question. Further, authors could use comparative genome analysis to provide hints as what genes could be involved (authors do have transcriptome dataset). I would anticipate that since *P. californicum* have STORR enzymes but does not produces morphinans, it should have either evolved or used a set of genes that are specific to this genome and probably absent to rest of the genomes described in this study. Is it possible to find those genes? For me, answering these questions will then provide an overview of how such an important event of STORR fusion directed the genome organization and thus caused or shaped the present-day chemotypic properties of the plant species.

In my opinion, estimation of STORR fusion event occurrence is interested, but then how that crucial event influenced the subsequent evolution of pathways leading to morphanes and other metabolites are not clear. This study fails to provide important insight into the evolutionary origins of genes involved in the biosynthesis of the target metabolites post STORR fusion events. There are so many interesting questions that have general implications to understand evolution of specialized metabolites in plants. I think that with the kind of data being generated in this study, authors need to mine data deeper to explore and discuss evolutionary aspect of BIAs biosynthesis. With the kind of resources authors have in this study, I feel disappointed that the authors have kept their focus too narrow, which is why I find it not suitable for Nature communication.

Minor comments-

1. In the method section, please provide the BUSCO program version, and what linages were used to estimate genome completion
2. Page 11, "We found that *P. bracteatum* shows very good synteny with the opium poppy regions containing the pro-morphinan genes with a notable difference being". Better to leave terms like "very good" from here.

Revision of Nature Communications manuscript NCOMMS-21-42033-T

Catania *et al.*, "A functionally conserved *STORR* gene fusion in *Papaver* species that diverged 16.8 million years ago"

Response to Reviewers Comments

We would like to thank the reviewers and the editor for all the constructive and positive feedback on our manuscript.

In the following text all reviewer comments are italicized, author responses are non-italicized, figures/tables citations are in bold and revisions in excerpts from main text are highlighted in red.

REVIEWER COMMENTS

Reviewer #1 (Remarks to the Author):

Thank you for following my suggestions.

*In Line 220 you call *P. californicum* and "Old World" species. As it only occurs in Americas, this should be a "New World species"*

We thank reviewer #1 for the comments on our revised manuscript, the text on page 12 has now been modified to clarify the 'Old World/New World':

"... *P. californicum* is a *Papaver* species indigenous to California ('New World') thought to have evolved in parallel to the Eurasian 'Old World' members of the family^{24,27,29,30}. The specific distribution of *P. californicum* to North Western America is an example of an 'Old World/New World' disjunction at 28 - 10 MYA in the Papaveraceae distribution³⁰. ..."

Reviewer #2 (Remarks to the Author):

*In the present revision for the manuscript, titled, "A functionally conserved *STORR* gene fusion in *Papaver* species that diverged 16.8 million years ago", authors have provided new analysis by including two of the previously published genomes and rebuttal for several of my comments. Authors in the current version of the manuscript reanalyzed the entire dataset by including previously published genomes of *P. rhoeas* and *P. setigerum*. In the rebuttal, the authors mentioned that using the previously published Poppy genome (which is higher quality in terms of assembly contiguity) does not change the main conclusion of this manuscript. I accept the argument. I also note the fact this study was published as BioRxiv prior to the Yang et al publication.*

We thank the reviewer for accepting our argument in response to the various points raised by the reviewer in the previous round.

*After going through all the responses from the authors and the changes that were made while addressing all reviewers, I now find that this study in its present form missed to address many interesting analyses that would offer deep insights on the evolution of specialized metabolites and the mechanism post the emergence of *STORR* fusion event. The *STORR* fusion event and its importance has been previously described in detail. The only core message from this study emerges as that the origin of the *STORR* gene fusion across the *Papaveraceae* family occurred between 16.8-24.1 million years ago before the separation of *P. californicum* from the other *Papaver* species. In that sense, I agree with reviewer #4 comments in the first round of review, and I am not satisfied with the explanation*

that authors have provided. With the kind of data authors have generated, I strongly feel that authors could find footprints of how STORR fusion events shaped the present-day existing biosynthesis pathway.

Without this, I find the correction of previously published Nature Communication article by incorporating more species is not enough.

One of my major criticisms of this manuscript as mentioned in several of my comments were the poor contiguity of the genome, for which, the author's rebuttal is as follows-

*"We agree with the reviewer that high contiguity in genome assemblies is essential for global comparison of genomes. However, our synteny analysis in Figure 4 focusses on the comparison of the region surrounding the STORR gene in different Papaver species and its relation to orthologs of other promorphinan genes of the opium poppy BIA gene cluster. We do not make any claims based on whole-genome assembly comparisons in this manuscript. The synteny that we do report in Figure 4 is based on single contigs and does not depend on the overall contiguity of the respective genome assemblies. For example, the *P. bracteatum* and *P. armeniacum* BIA cluster regions shown in Figure 4a are based on evidence from single contigs. Improving the genome assembly would have little or no effect on the quality of the evidence presented in Figure 4."*

For me, this then becomes a very specific question and I then find this article more suitable to a plant specific journal and not within the broad readership of Nature communication. This study will then be of interest to only those researchers who are interested in papavers species, especially using the generated data resource for further functional analysis. Other than the fact that we improved the estimated time of the evolution of STORR, I fail to see how the results are applied to explore the overall evolution of the biosynthesis pathways based on this knowledge.

We assume here that the other reviewer being referred to by reviewer #2 is reviewer #3 rather than reviewer #4. We have argued in our previous response to reviewer #3 that our findings are significant to understand the origin of the STORR gene fusion and STORR function across the Papaveraceae family. Our discovery of the presence of (*R*)-glaucine in *P. californicum* and association with STORR lays the foundation to elucidate the biosynthetic pathway of these important compounds. Our manuscript is not just limited as a correction of a previously published Nature Communications article, even though that in itself is also very important for the advancement of science.

My reason for advocating on improving the genome assemblies was with the hope that a comprehensive comparative genome analysis could reveal several questions such as-
1. Does the evolution of STORR result in the loss or gain of a specific set of enzymes that are involved in the biosynthesis pathway of core metabolites

We thank reviewer #2 for raising this point. Yes, our analysis has suggested that the neo-functionalization and clustering of the promorphinan genes in the *Papaver* species occurred after the evolution of STORR, as described in **Figure 4**. Orthologues of SALSYN, SALAT, SALR and THS have only been identified in the genomes of a subset of the Clade 2 *Papaver* species that produce promorphinan/morphinan compounds and contain the STORR gene fusion (**supplemental table S5**). We have now calculated the percentage identity matrix both at nucleotide and protein level for orthologues of SALSYN, SALAT, SALR and THS in these species. We have appended these results to **supplementary datasets 10&11 and supplemental table S7**. These values range from 94.1-99.8% and 90.5-100% respectively, also supporting the case that the evolution of the promorphinan/morphinan biosynthetic pathway was triggered by the STORR gene fusion event.

We have modified the following text to provide more clarity on this point on page 11:

“Orthologues of *SALSYN*, *SALAT*, *SALR* and *THS* have only been identified in the genomes of a subset of the Clade 2 *Papaver* species that produce promorphinan/morphinan compounds and contain the *STORR* gene fusion (supplemental table S5). Percentage nucleotide and amino acid identity matrix of these orthologues range from 94.1-99.8% and 90.5-100% respectively (supplemental table S7, supplementary datasets 10&11), supporting the case for evolution of the promorphinan/morphinan biosynthetic pathway being triggered after the *STORR* gene fusion event.”

We have also revised **Figure 4a and 4b** to remove one extra copy of *SALAT* in the *P. setigerum* chromosome 8 cluster after a close inspection of the annotations from the Yang *et al* publication. We found that two genes, Pse08G45690.0 and Pse08G45700.0, were mis-annotated (supplementary table S9) by Yang *et al*, since in fact they represent two fragments of a continuous intronless *SALAT* pseudogene, which contains one frameshift and one nonsense codon mutation in the open reading frame. Based on our analysis of the genomic DNA alignment of the *STORR* genes, the *STORR* copy Pse08G45740.0 in the *P. setigerum* chromosome 8 cluster is also likely to be a pseudogene as it contains an additional insertion in the open reading frame besides the introns at the conserved positions.

We have modified the following text on pages 9-10 of the main text:

“... Both syntenic regions in *P. setigerum* contain *STORR* and all four promorphinan genes. Two annotated genes, Pse08G45690.0 and Pse08G45700.0, showing similarity to *SALAT* are arrayed in tandem in the syntenic region on *P. setigerum* Chromosome 8 (supplemental table S9). Close inspection using sequence alignment indicates that both Pse08G45690.0 and Pse08G45700.0 are mis-annotated in the reported annotation¹⁴ and instead represent two fragments of a continuous intronless *SALAT* pseudogene, containing one frameshift and one nonsense codon mutation in the open reading frame. Nucleotide sequence alignment of *STORR* genomic DNA regions suggests that the *STORR* copy Pse08G45740.0 in the *P. setigerum* chromosome 8 cluster is also likely to be a pseudogene as it contains an additional insertion in the open reading frame besides the introns at the conserved positions. ...”

Reviewer #2 also asks if there is gene loss triggered by the *STORR* fusion event. Gene loss in a genome can be caused by many mechanisms for example during rearrangements following whole genome duplications. While we are able to present solid evidence attributing the *STORR* gene fusion event as a trigger for the evolution of the promorphinan/morphinan biosynthetic pathway by neofunctionalization in *Papaver* species, we are much less confident in making causal link statements about the *STORR* gene fusion event and specific gene loss.

2. Clearly *P. californicum* has functional *STORR*, but it does not produce several of the core metabolites that Poppy does. Why? Does it have to do with the gain or loss of some specific genes? A comprehensive genome-wide synteny analysis could provide such answers.

We assume here that when the reviewer refers to core metabolites they mean the promorphinans as shown in **Figure 1b**. The reason why *P. californicum* does not accumulate these compounds is because it does not contain any of the genes associated with their biosynthesis from reticuline. *P. californicum* represents the earliest branching species in the *Papaver* lineage that we have found to contain the *STORR* gene fusion which, based on our data, happened between 24.1 - 16.8 MYA. While we are confident that the *STORR* gene fusion event was before the neofunctionalization of promorphinan genes we cannot conclude that the latter was before or after the branching of *P. californicum* from the other Clade 2 species as shown in **Figure 1**. Therefore it is possible that either neofunctionalization did not happen prior to the branching of *P. californicum* or it did happen and the genes were lost. Genome-wide synteny analysis is not likely to shed any further light on this matter.

We have modified the text on page 11 to reflect this discussion point as follows:

“*P. californicum* represents the earliest branching species in the *Papaver* lineage that we have found to contain the *STORR* gene fusion event occurring between 24.1 - 16.8 MYA (Fig. 1c). While we are confident that the *STORR* gene fusion occurred before the neofunctionalization of promorphinan genes we cannot conclude whether the latter was before or after the branching of *P. californicum* from the other Clade 2 species as shown in Figure 1. Therefore it is possible that either neofunctionalization did not happen prior to the branching of *P. californicum* or it did happen and the genes were lost.”

3. Does enzymes known to be involved in the pro-morphinan/morphinan subclass of BIAs biosynthesis followed the event of *STORR* fusion. Authors reported the rate of substitution in their previously published article on Poppy genome, and they could estimate an approximate time of emergence of key genes and judge their emergence with *STORR*.

4. Knowing the approximate timing of *STORR*, what enzymes were evolved or retained or gained around that time period within producing and non-producing species, does that has any influence on the evolution of the core metabolites?

Points 3 and 4 overlap with the issues raised in Points 1 and 2, which we have comprehensively addressed above.

5. Does the copy number of genes involved in the pro-morphinan/morphinan subclass of BIAs biosynthesis increase within producing plants compared to the ones that does not produces?

This information is provided in **Figure 4**. Those *Papaver* species investigated that did not produce the promorphinan/morphinan subclass of BIAs also do not contain any of the genes involved in their synthesis. In those that do produce pro-morphinan/morphinan alkaloids the respective genes required for the synthesis of promorphinans up to the first morphinan alkaloid, thebaine, are clustered with *STORR* in the morphinan gene cluster. The genome of *P. bracteatum* contains one copy of this morphinan gene cluster. The genomes of *P. somniferum* and *P. setigerum* each contain two copies of this morphinan gene cluster. In opium poppy one copy of the morphinan gene cluster has lost the *STORR* gene. *P. armeniacum* has lost three genes of the morphinan gene cluster, retaining only *STORR* and *SALSYN*. Consistent with this finding *P. armeniacum* does not produce promorphinan alkaloids beyond Salutaridine and consequently no morphinan alkaloids as shown in **Figure 1b**.

It should be noted that an increase in copy number of pathway genes is only one way to increase production of specific metabolites. Thus, increased copy number would be at best an indication that flux through the pathway may be increased by increasing overall enzyme abundance/activity. The same could be achieved by other mechanisms such as an increase in gene expression that does not necessarily rely on or require multiple gene copies.

6. Does the event of *STORR* fusion had any role in shaping or reorganizing genomes of the producing plants?

7. Does plants producing pro-morphinan/morphinan show high degree of collinearity overall? How about the gene clusters?

Our work has focused on understanding how the *STORR* gene fusion event has impacted on the subsequent biosynthesis of novel classes of BIAs in plants. Our experimental design does not address questions relating to overall collinearity or chromosomal rearrangement across the *Papaver* lineage and how that might relate to specific genes such as *STORR*. To address such questions would require more genome assemblies of closely related Clade 1

Papaver species that diverged immediately prior to *STORR fusion* in order to establish if there is any correlation between the *STORR fusion* event and local or global genome re-organisations.

8. As authors mentioned the fusion event of *STORR* does not mean that the plant will produce pro-morphinan/morphinan, then what is the determining factor? Is there an essential set of genes that one could propose? I am not sure if such a method exists, the fact remains that the *STORR* genes require key precursors for subsequent reactions downstream. Therefore, by comparing *P. californicum* with rest of the producing plants, and by including the plants which observed no fusion of *STORR*, one could find genes that evolved under influence of *STORR* fusion and a set of genes that further gained to offer present-day chemotypes.

These and several additional analyses and answers could be derived using the dataset that authors have generated in this study. In my opinion, the goal of this study needs to be broadened.

Authors did report (R)-glaucine in *P. californicum*, and therefore, its genome becomes valuable. However, that's not the main question. Further, authors could use comparative genome analysis to provide hints as what genes could be involved (authors do have transcriptome dataset). I would anticipate that since *P. californicum* have *STORR* enzymes but does not produce morphinans, it should have either evolved or used a set of genes that are specific to this genome and probably absent to rest of the genomes described in this study. Is it possible to find those genes? For me, answering these questions will then provide an overview of how such an important event of *STORR fusion* directed the genome organization and thus caused or shaped the present-day chemotypic properties of the plant species.

In my opinion, estimation of *STORR fusion* event occurrence is interesting, but then how that crucial event influenced the subsequent evolution of pathways leading to morphanes and other metabolites are not clear. This study fails to provide important insight into the evolutionary origins of genes involved in the biosynthesis of the target metabolites post *STORR fusion* events. There are so many interesting questions that have general implications to understand evolution of specialized metabolites in plants. I think that with the kind of data being generated in this study, authors need to mine data deeper to explore and discuss evolutionary aspect of *BIA*s biosynthesis. With the kind of resources authors have in this study, I feel disappointed that the authors have kept their focus too narrow, which is why I find it not suitable for Nature communication.

Point 8 re-visits the various other points raised by reviewer #2. Our responses to those other points describe why we strongly disagree with this analysis of our manuscript. For example, the statement that '*This study fails to provide important insight into the evolutionary origins of genes involved in the biosynthesis of the target metabolites post STORR fusion events.*' is simply not consistent with the data and conclusions we present.

Furthermore, our manuscript corrects the recent literature that wrongly associates the *STORR* gene fusion event and subsequent evolution of morphinan biosynthesis with a whole genome duplication event that occurred in *P. somniferum* 7 - 8 MYA.

Minor comments-

1. In the method section, please provide the BUSCO program version, and what lineages were used to estimate genome completion

The BUSCO program version 4.14 and lineage embryophyta_odb10 was used to estimate genome completion. These details along with the accompanying reference have been updated in the methods section on page 18 and references page 27:

“...Functional domains of protein-coding genes were identified with InterProScan version 5.52-86 with default parameters. BUSCO analysis was run on the finalised annotations of the

five *Papaver* genomes using BUSCO version 4.1.4 and embryophyta_odb10 used to estimate genome completion⁶⁵.”

2. Page 11, “We found that *P. bracteatum* shows very good synteny with the opium poppy regions containing the pro-morphinan genes with a notable difference being”. Better to leave terms like “very good” from here.

The relevant text on page 10 has been modified to remove the word ‘very’

Reviewers' Comments:

Reviewer #2:

Remarks to the Author:

In the second revision, authors have provided point-to-point rationale and have addressed most of my concerns related to the wider application of such an interesting study design and good quality dataset. I agree with all these comments and admire their patience to further improve this manuscript from the previous version. Certainly, additional analysis does help to improve the science, and the fact that results from this study does provide a better over-view of how genome evolved and shaped the present BIA biosynthesis that we are aware of and its association with STORR gene fusion event. In that sense, the knowledge of when this happened does provide further insights and will be important to explore biosynthesis of important metabolites.

I am left with a two last comments, and despite of feeling the fear of being judged as too difficult reviewer, I feel that these changes will improve the use of the produced datasets in this study.

1. I leave it to the authors description, but if they could make the databases (all assembled genome, comparative genome analysis, genome viewers, synteny datasets, expression datasets including FPKM or count data) in form of a dedicated website, that will be a great service to the plant biologist working in this feild. It could be in the same line as Solnet (<https://solgenomics.net/>) or Plant genome garden (<https://plantgarden.jp/ja/index>) or similar types. If I am not wrong, there are communities that are eager to host such datasets, and they will be very happy to offer their website or to provide template to host such datasets. Again, just a suggestion.

2. Please provide a git-hub or similar ways of sharing all the scripts used for analysis. This article includes interesting interpretations using comparative genome analysis, phylogenomics together with genome assembly and other methods to conclude the results. If the script used in this study are available, that will be a great help to a wider audiences.

I again want to thank authors for their hard work and for the patience to address most of my concerns.

Revised Nature Communications manuscript NCOMMS-21-42033-B

Catania *et al.*, "A functionally conserved *STORR* gene fusion in *Papaver* species that diverged 16.8 million years ago"

In the following text all reviewer comments are italicized, author responses are non-italicized.

Finally, we would like to thank all reviewers and the editor for their constructive and positive feedback on our manuscript.

Response to reviewer's comments:

REVIEWER COMMENTS

Reviewer #2 (Remarks to the Author):

*In the second revision, authors have provided point-to-point rationale and have addressed most of my concerns related to the wider application of such an interesting study design and good quality dataset. I agree with all these comments and admire their patience to further improve this manuscript from the previous version. Certainly, additional analysis does help to improve the science, and the fact that results from this study does provide a better over-view of how genome evolved and shaped the present BIA biosynthesis that we are aware of and its association with *STORR* gene fusion event. In that sense, the knowledge of when this happened does provide further insights and will be important to explore biosynthesis of important metabolites.*

We thank the reviewer for accepting our arguments and positive feedback on the revised manuscript.

I am left with a two last comments, and despite of feeling the fear of being judged as too difficult reviewer, I feel that these changes will improve the use of the produced datasets in this study.

1. I leave it to the authors description, but if they could make the databases (all assembled genome, comparative genome analysis, genome viewers, synteny datasets, expression datasets including FPKM or count data) in form of a dedicated website, that will be a great service to the plant biologist working in this feild. It could be in the same line as Solnet (<https://solgenomics.net/>) or Plant genome garden (<https://plantgarden.jp/ja/index>) or similar types. If I am not wrong, there are communities that are eager to host such datasets, and they will be very happy to offer their website or to provide template to host such datasets. Again, just a suggestion.

2. Please provide a git-hub or similar ways of sharing all the scripts used for analysis. This article includes interesting interpretations using comparative genome analysis, phylogenomics together with genome assembly and other methods to conclude the results. If the script used in this study are available, that will be a great help to a wider audiences.

I again want to thank authors for their hard work and for the patience to address most of my concerns.

We thank the reviewer for these suggestions on both data and script sharing. All data and methods associated with this manuscript can be easily accessed as described in the 'Data Availability' section of the manuscript and the 'Reporting Summary' file.